# On the link between conscious function and general intelligence in humans and machines

**Arthur Juliani**                                          *ajuliani@microsoft.com*
*Microsoft Research. New York City, USA*
*Araya Inc. Tokyo, Japan*

**Kai Arulkumaran**                                   *kai_arulkumaran@araya.org*
*Araya Inc. Tokyo, Japan*

**Shuntaro Sasai**                                   *sasai_shuntaro@araya.org*
*Araya Inc. Tokyo, Japan*

**Ryota Kanai**                                          *kanair@araya.org*
*Araya Inc. Tokyo, Japan*

**Reviewed on OpenReview:** *https://openreview.net/forum?id=LTyqvLEv5b*

## Abstract

In popular media, there is often a connection drawn between the advent of awareness in artificial agents and those same agents simultaneously achieving human or superhuman level intelligence. In this work, we explore the validity and potential application of this seemingly intuitive link between consciousness and intelligence. We do so by examining the cognitive abilities associated with three contemporary theories of conscious function: Global Workspace Theory (GWT), Information Generation Theory (IGT), and Attention Schema Theory (AST). We find that all three theories specifically relate conscious function to some aspect of domain-general intelligence in humans. With this insight, we turn to the field of Artificial Intelligence (AI) and find that, while still far from demonstrating general intelligence, many state-of-the-art deep learning methods have begun to incorporate key aspects of each of the three functional theories. Having identified this trend, we use the motivating example of mental time travel in humans to propose ways in which insights from each of the three theories may be combined into a single unified and implementable model. Given that it is made possible by cognitive abilities underlying each of the three functional theories, artificial agents capable of mental time travel would not only possess greater general intelligence than current approaches, but also be more consistent with our current understanding of the functional role of consciousness in humans, thus making it a promising near-term goal for AI research.

## 1 Introduction

In popular media, there is often a link drawn between the advent of consciousness in artificial agents and those agents exhibiting human or even superhuman level intelligence. We see such a connection again and again in science fiction films such as 2001: A Space Odyssey, Ghost in the Shell, Ex Machina, or Her. In each case, self-awareness on the part of the artificial agent co-occurs with that agent's ability to outsmart or manipulate the humans they interact with. Despite its prevalence in media, the validity of this apparent connection is often neither critically examined, nor explained within the context of the media itself. Here we explore whether this link is more than just intuitive by looking at the supporting role that conscious experience may play in both human and artificial intelligence.

We start with a scientific perspective on the question of consciousness and first-person experience. While still controversial, contemporary theorists believe that consciousness can be studied as an evolutionarily developed trait found in animals (Dennett, 2018). From this perspective, consciousness can be associated with a certain set of adaptive behavioral functions, and can be examined scientifically with respect to these functions. While the exact nature of these functions, and their relationship to consciousness is still widely argued, the past few decades has seen the field mature, and a set of theories gain relatively wider acceptance, enjoying an accumulation of both theoretical and empirical support.

We start with a common framework for understanding the function of consciousness: information access (Block, 1995). Unlike "phenomenal consciousness" which corresponds to any subjective experience, regardless of how subtle, "access consciousness" corresponds to information one is aware of experiencing, and can thus report. We examine three well known contemporary theories of the functional basis of consciousness from the perspective of conscious access: Global Workspace Theory (GWT) (Baars, 1994; 2005), Information Generation Theory (IGT) (Kanai et al., 2019), and Attention Schema Theory (AST) (Graziano & Webb, 2015). Each of these were chosen for their unique set of theoretical bases and predictions concerning conscious access.

The Global Workspace Theory provides an account of the function of attention, working memory, and information sharing between brain modules in humans and other mammals, and has garnered a body of neuroscientific work which supports it (Mashour et al., 2020). The Information Generation Theory provides an account of the internal generation of trajectories of coherent experience in humans, and is based on clinical and research findings concerning the neural basis of consciousness. Finally, Attention Schema Theory provides an account of flexible adaptation to novel contexts by an attentional system, and is based on a number of pieces of evidence from cognitive neuroscience literature. Each of these theories both provides a behavioral role for consciousness as well as describe a mechanism whereby intelligent, adaptive behavior may be made possible.

How various cognitive abilities might contribute to intelligence depends on how one chooses to define intelligence, a term with a varied and complicated history (Legg & Hutter, 2007). Utilizing the framework proposed by Francois Chollet (2019), we define intelligence as the ability to quickly acquire novel skills with little direct experience, knowledge, or structural priors related to the skills being acquired. Put another way, if two agents can learn to solve a new task to the same proficiency level, but one solves it with less experience, knowledge, or in-built structural advantage than the other agent, then we say that the former agent is the more intelligent of the two.

The appeal of this definition is that it remains largely agnostic to anthropocentric trappings of defining intelligence with respect to specific complex skills, such as tool or language use. Humans for example may benefit from hands or vocal cords which enable specific kinds of tool use or vocalizations that other animals are not capable of. On the other hand, these specific cases can be seen as resulting from the more general definition, with language and tool use both being examples of skills which require variable amounts of experience or prior knowledge to acquire. In the case of chimpanzees for example, sign language use can be acquired, but at a much slower rate than that of humans, suggesting that in this domain at least, they are less generally intelligent (Gardner & Gardner, 1969).

The study of intelligence can naturally be extended beyond biological agents to artificial ones. Much of the current research in the field of Artificial Intelligence (AI) seeks to develop agents which display human or superhuman levels of intelligence (Lake et al., 2017; Mnih et al., 2015; Vinyals et al., 2019). Using the definition above however, many contemporary artificial agents do not qualify as being particularly intelligent when compared to many animals. Even state-of-the-art AI systems for image recognition or game playing often require multiple orders of magnitude more prior experience or pre-defined domain knowledge in order to solve these tasks than do humans or even other animals (Badia et al., 2020; Dai et al., 2021). Even with this knowledge or experience, these models are often limited in their ability to adapt to new tasks after previously learning similar ones. For example, a model trained to play a specific game within the Arcade Learning Environment (Bellemare et al., 2013) is only able to perform at chance level when exposed to different tasks within the same game (Kansky et al., 2017). As such, there exists an intelligence gap, one

which can be broken down into various cognitive abilities, and the systems which may or may not support them.

When examining the three proposed theories of conscious function, we find that each describes a system which supports greater domain-general skill acquisition, and thus greater intelligence by our adopted definition. Utilizing this insight, we turn to current state-of-the-art AI systems, and find that systems which utilize specific aspects of each theory can result in greater generalization than other similar models (Hendrycks et al., 2021). Other prominent lines of research to address limitations in general intelligence of artificial agents include research into modularity and causality, learning world models, and meta-learning. Each of these can be seen as partially mapping onto one of the three functional theories of consciousness explored here.

Given the limited nature of generalization in these artificial systems, and the strong connection between approaches to improving generalization and the functional theories of consciousness laid out, we turn to a motivating example to describe how these disparate lines of research can be made to converge. To do so, we examine the phenomenon of mental time travel (Tulving, 2002), a cornerstone of human memory and imagination. As defined by Endel Tulving, mental time travel is the ability to project oneself into the past or future and actively participate in a series of imagined events within the projection. While controversial, this ability has been argued to be both uniquely human and specifically conscious (Suddendorf et al., 2011).

One can contrast mental time travel as described by Tulving with the phenomenon of experience replay found in other animals and its algorithmic equivalent in artificial agents (Lin, 1992; Mnih et al., 2015; Foster, 2017). Because it involves only re-experiencing past events, experience replay results in agents whose ability to adapt to novel scenarios is markedly limited. We make the further distinction between mental time travel and experience preplay, which consists of the simulation of future experiences within a limited environmental and behavioral context (Pezzulo et al., 2014; Ólafsdóttir et al., 2015). Mental time travel in contrast requires the generation of experience trajectories from possible environments and possible policies, a much broader distribution.

Taking stock of these differences in functional properties and generalization ability, we finally propose criteria by which artificial agents which are capable of true mental time travel might be developed. By creating systems which can utilize selective attention (Posner, 1994; Ganeri, 2017), represent multi-modal structured information in a domain-agnostic fashion (as in GWT), generate spatially and temporally coherent imagined trajectories of experience (as in IGT), and adapt the attention policy governing information access to the problem at hand (as in AST), mental time travel as described by Tulving can be made possible. In doing so, we expect that artificial agents with the ability to mentally time travel will almost certainly also display greater general intelligence than those found today, making them a promising goal for AI research in the near term.

The rest of the text is structured as follows. Section 2 provides a background on the scientific study of consciousness and the difference between phenomenal and access consciousness. Readers with a strong background in consciousness studies can feel free to skim over this section. Section 3 then turns to describing the three contemporary theories of conscious function under consideration here: GWT, IGT, and AST. Next, Section 4 turns to the question of intelligence in animals and artificial agents. It is followed by Section 5, which provides a survey of approaches in the field of AI to achieve more general intelligence. Readers with a strong background in AI research can feel free to skim over this section. In Section 6, we propose mental time travel as a cognitive phenomenon which is uniquely useful for understanding the link, and distinguish it from related phenomena of replay and preplay. In Section 7, we discuss potential ways to unify the three theories of conscious function, as well as how such a system might be implemented in artificial agents. Finally, in Section 8 we return to the motivating question of the paper, and provide a provisional answer. The last three sections comprise the main theoretical contributions of this work, and we hope their contents are not only of interest to both consciousness and AI researchers, but will help to contribute to increasing dialogue between the two fields.

## 2  Science of consciousness

In order to make sense of the link between consciousness and intelligence, we must first define more clearly what exactly is meant by consciousness. We use as our starting point the definition proposed by Thomas Nagel: consciousness is "what it feels like to have an experience" (Nagel, 1974). The specific nature of this "feels like" at any given moment is often referred to as qualia, a reflection of the qualitative nature of first-person experience (Kanai & Tsuchiya, 2012). Historically the field of phenomenology has attempted to provide a theory of qualia within its own terms, that is to say from a qualitative and subjective rather than objective perspective (for early 20th century examples, see Husserl (1970) or Merleau-Ponty (2013)). Despite the wealth of insights developed from phenomenology, a subjective definition of consciousness proposes challenges for more rigorous scientific inquiry.

In an attempt to make the study of consciousness more amenable to such scientific inquiry, the question of understanding consciousness at large proposed by Nagel and earlier phenomenologists was divided into two broad subsets of problems by David Chalmers (1995). They were grouped into the so-called easy problems and hard problem of consciousness. The easy problems concern understanding the neural correlates and content of consciousness. These problems are amenable to the typical scientific approach, where specific hypotheses can be proposed, experimentally tested, and nullified. Such a hypothesis could be: "Is the prefrontal cortex necessary in order to report the content of visual perception in humans." That hypothesis can then be tested through either the collection of observational clinical data or experimental intervention, and the hypothesis could be nullified if deactivation of the region does not produce a measurable change in conscious experience on the part of the subject.

In contrast, the hard problem of consciousness is defined as the problem of understanding why a specific configuration of matter (the brain) is associated with a specific subjective experience (qualia). While various experimental paradigms might explain what the relationship between matter and qualia is, there is no such paradigm which would provide insight into why that relationship exists to begin with. In a sense the easy problems are problems of what and how, and the hard problem is the problem of why. With this distinction in place, empirical scientists could begin to focus on the questions in the former category, while philosophers and theoreticians could continue to approach the latter.

A further distinction was made by Daniel Dennett (2018) who proposed a slightly different "hard question" to complement the hard problem. In contrast to the hard problem of "why is there consciousness?" Dennett asks "what is consciousness for?" This question once again takes the problem of consciousness from a purely philosophical domain and grounds it in the biological and evolutionary. In doing so, it provided additional avenues through which consciousness could be amenable to scientific inquiry, specifically on functional grounds. More relevant to the topic of this work, asking about the potential biological function of consciousness means asking about the ways in which consciousness may support adaptive behavior in living organisms. Asking about adaptive behavior naturally leads to questions concerning cognition and thus intelligence.

By starting with a functional perspective on consciousness, one can begin to make distinctions between the qualitative nature of qualia and the more quantifiable set of information that is or is not accessible to conscious experience at any given moment. This distinction was made formally by Ned Block (1995), who proposed the term phenomenal consciousness for the former, and access consciousness for the latter. Importantly, the properties of access consciousness explicitly relate to conscious function since they are biologically conditioned and can be directly measured. According to Block, this includes information which is "available for use in reasoning and rationally guiding speech and action," and as such, can be alternatively described as information which is available for cognitive access, without recourse to a discussion of qualia or phenomenology (Block, 2008). This latter interpretation will be of importance when considering the potential for conscious (or "cognitive") access in artificial agents for which there is no ground for assuming phenomenal consciousness. The study of conscious information access also overlaps with the broader field of cognitive science, in its study of attentional capacity, control, and working memory, all three of which are also core to intelligence (Schweizer et al., 2005).

Just as the nature of access consciousness suggests a potential functional role in the behavior of an organism, phenomenal consciousness by definition precludes such a role. This reality was pointed out by Chalmers in

his famous thought experiment concerning what he referred to as philosophical zombies (Chalmers, 1996). In the thought experiment, Chalmers asks us to imagine so-called zombies which are able to act in ways which are indistinguishable from normal conscious humans, except for the fact that they lack any form of phenomenal experience. While this concept strikes some as difficult at first to accept, there is nothing which necessarily precludes such beings from existing, at least hypothetically. On the other hand, it is difficult to imagine beings which would act exactly like normal humans but lacked the ability to access and maintain information over time in the way described by the construct of access consciousness. Inversely, we can imagine beings capable of rich phenomenal experiences, but which neither reason based on this experience, nor turn that experience into action, let alone intelligent action. As such, the link proposed here between intelligence and consciousness concerns exclusively access consciousness and the proposed functional roles of consciousness.

We do not however totally discount the possibility that the study of phenomenal consciousness might provide functional insights into animal behavior. For example, recent work by Mark Solms has suggested that the specific affective valence of a given qualia may have implications for evaluative behavior (Solms, 2021). We leave the study of such implications to others, and here focus on access consciousness as defined by Block, and the relevant functional theories which can be interpreted within that framework. There are also other popular theories of consciousness not examined below, such as Integrated Information Theory (IIT), which is a theory of phenomenal consciousness. IIT applies to the organization of any system, regardless of that system's behavioral repertoire (Tononi et al., 2016). As such, it generally does not make specific functional hypotheses regarding the systems which may or may not be conscious, and thus has limited relevance for the questions here concerning intelligence.

## 3 Possible functions of consciousness

Given that the dynamics of what content makes it into consciousness plays some functional role in behavior, we can begin to explore the current field of theoretical work which proposes various explanations for what that role is. In particular, we examine the Global Workspace Theory (GWT), Information Generation Theory (IGT), and Attention Schema Theory (AST), three popular contemporary accounts of the function of consciousness. In addition to grounding consciousness from a functionalist perspective, each theory also proposes brain networks and dynamics which are hypothesized to be involved in the relevant computation. As such, each theory is amenable to scientific scrutiny, and indeed each is supported by at least some neuroscientific findings.

Before describing the particulars of each theory, it is worth discussing a more general approach to understanding conscious access which we take as a starting point. Key to conscious access is an understanding of attention. We define attention broadly as the selection of some subset of potentially available information and the necessary exclusion of other subsets, or as James (2007) put it "...the taking possession by the mind, in clear and vivid form, of one out of what seem several simultaneously possible objects or trains of thought." This so-called attentional perspective has been proposed within the domain of cognitive science by Posner (1994), and the domain of philosophy of mind by Ganeri (2017). Attention applies not only to the traditional sensory modalities, but also to internal body signals, thought, memory, and affect.

Perhaps unsurprisingly, the importance of attention follows directly from the definition of access consciousness given above. Whereas at any given moment the potential phenomenal experience can be quite expansive, what we as conscious agents actually experience is in any given moment limited. This is because there exists in all known conscious agents a mechanism by which certain information makes it into consciousness and other information does not. We say that we have attended to, and are thus aware of, information which becomes conscious, and are unaware of or inattentive to that which does not. It is important to note however that despite their close coupling, the awareness which derives from conscious access and the psychological notion of attentional control are not equivalent, and can be dissociated in various controlled contexts (Koch & Tsuchiya, 2007). This potential dissociation has important implications for the theories considered here, especially AST.

Block summarizes this phenomenon of awareness with the term "perception overflows access" (Block, 2011). In doing so, he uses the metaphor of a lottery, whereby many pieces of proto-conscious information could

potentially win the lottery and become consciously accessible, but only some do. This phenomenon has been empirically validated in experimental settings, where participants are shown more information than they are able to access (and thus report), while still phenomenally "experiencing" that information. What enters consciousness and what does not however is not random, but rather behaviorally guided, through evolution, learning, endogenous, and exogenous forces.

It is worth noting that the distinction between phenomenal and access consciousness is not universally accepted, and in some cases has been questioned by the proponents of the theories presented below (Dehaene & Naccache, 2001; Graziano et al., 2020). Despite this disagreement, and in order to provide a unified basis for analyses, we propose that each of the three theories discussed below can still be understood as explanations for how and why at any given moment certain information enters consciousness and other information does not. As such, each theory can be viewed from the lens of access consciousness, as defined by Block (1995). Below we provide a description of each theory, and discuss its implication for conscious information access. For a schematic overview of the three theories, see Figure 1.

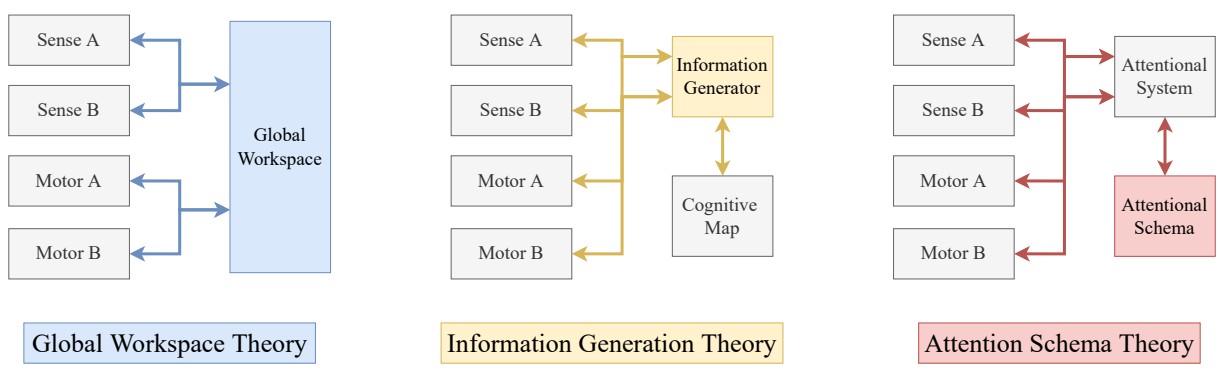

Figure 1: Schematics of different functional theories of consciousness. Highlighted boxes correspond to the source of purported content of consciousness for each theory. **Left**: Global Workspace Theory. Various modules, including sensory and motor modules interact via a shared global workspace which information is both broadcast to and from. Conscious awareness corresponds to the contents of the workspace. **Middle**: Information Generation Theory. Sensory and motor modules contribute to a generative model of the dynamics of the agent in the environment. The spatial and temporal dynamics of the generative model are governed by a cognitive map which ensures coherence. The predictions of the generative model are what make it into conscious awareness. **Right**: Attention Schema Theory. A high-level model of the attentional process is generated and interacts with the attentional system. Conscious awareness corresponds to the state of this model.

## 3.1 Global Workspace Theory

The first theory we consider is the Global Workspace Theory (GWT), and its updated formation, the Global Neuronal Workspace Theory (Baars, 1994; Dehaene et al., 1998; Baars, 2005). For simplicity, we refer to both as GWT. This theory can be seen as naturally developing from an attempt to provide a formal account of cognitive information access. If there is only a limited amount of information which can be consciously experienced at a given moment, then there must be a mechanism by which that information is selected, maintained, and shared with the rest of the system. The global workspace proposed by Baars and others is just such a mechanism. The workspace is defined as a common representational space of fixed capacity where various modules within the brain can share information with other modules.

In its various formulations, this workspace has been associated with the frontal and parietal cortex and functionally connected to both attentional control and working memory (Dehaene et al., 1998; 2006). Attentional control can be interpreted as the specific policy for admitting behaviorally-relevant information into the workspace at the exclusion of irrelevant information. Working memory can be seen as the capacity to maintain information within the workspace for an extended period of time, as well as to manipulate the con-

tents of the workspace to meet behavioral goals (Mashour et al., 2020). There is also experimental evidence to suggest largely overlapping neural mechanisms responsible for these two functions, suggesting a more fundamental shared function, which the global workspace purports to describe (Panichello & Buschman, 2021). In addition to evidence suggesting that there are indeed central information processing regions of the brain, there is also evidence which explicitly supports the GWT as a model of cognitive access. Under certain circumstances, information is not consciously perceived unless there is sufficient activation in the prefrontal region, thus resulting in an ignition event, where that information is not only maintained, and thus consciously perceived, but is then also broadcast to other modules within the brain (Van Vugt et al., 2018).

## 3.2 Information Generation Theory

The second theory we consider is the Information Generation Theory (IGT), recently proposed by (Kanai et al., 2019). According to this theory, information which is consciously accessible is not simply first-order sensory input, but is rather always the result of a generative model within the brain, which during waking experience is most often conditioned on sensory input. This generative model is also capable of information generation which is contrary to the sensory input, or even completely disconnected from it, as is the case when dreaming. As such, the generation process confers the ability to not only predict the sensory world, but to simulate unseen events in that world. According to IGT, it is this simulation which corresponds to what is consciously accessible.

IGT can be linked to other similar descriptions of conscious experience derived from the expectations of Hierarchical Predictive Coding in the brain (Friston & Kiebel, 2009), Predictive Processing as described by Clark (2012), or other theories of top-down recurrent processing (Lamme & Roelfsema, 2000). In each case, the cortex is understood as largely being responsible for the generation of predictions concerning sensory or cognitive information, rather than merely representing such information directly. This process of prediction generation thus enables the framing of the cortex as a generative model which produces these predictions in response to sensory and motor contingencies. As such, it is these predictions which we are consciously aware of, not the incoming sensory information itself.

The IGT was derived from evidence for the dissociation between conscious perception and behavior which can sometimes be made apparent in individuals with various forms of brain damage. An example of this can be seen in experiments demonstrating form-agnosia, such as the mail slit experiment. In this work, researchers examined the ability of an individual who was phenomenally unconscious of their visual experience due to sustained brain damage to correctly put a piece of mail through a slot which could vary in its width or orientation. Despite being unable to report conscious awareness of any visual content corresponding to the mail or the slit, the individual was nevertheless able to accurately complete the task (Goodale et al., 1991). When the visual stimuli of the mail slip was removed before being asked to perform the corresponding action however, the same individual failed to complete the task, suggesting that a maintained model of the world is missing alongside the conscious experience of the stimuli (Goodale et al., 1994). The individual lacked the ability to generate and maintain information corresponding to their visual experience, and was likewise not consciously aware of that very information as well.

Alongside the ability to generate and maintain a model of the world is the ability to utilize this model in the simulation of experience over time. This capacity has elsewhere been referred to as the "experience construction system" of the brain, and been localized to the medial temporal lobe (Hassabis et al., 2007; Hassabis & Maguire, 2009), which is also the purported cite of the "cognitive map" used to ground navigation in the world (O'keefe & Nadel, 1978). Such a capacity for temporally extended experience generation is likely the result of an extended network of brain regions, involving frontal, parietal, and temporal connectivity (Preston & Eichenbaum, 2013). Given the temporally extended nature of conscious experience, we include the cognitive map as part of the large information generation system originally described in (Kanai et al., 2019).

### 3.3   Attention Schema Theory

The third theory we consider is the Attention Schema Theory (AST) (Graziano & Webb, 2015). According to AST, it is not attention which makes conscious access possible, but rather the capacity to represent a high-level model of the attentional process. In this view, the content of consciousness corresponds to the information present in the brain's model of attention, not the information being attended to itself. According to the proponents of AST, consciousness, including the feeling of having a subjective experience, is simply the result of this mental model, which they refer to as awareness, and distinguish from attention (Graziano et al., 2020). The functional role of the attention schema is then as a higher level controller responsible for monitoring and adapting the dynamics of the attentional process based on the particular behavioral needs of the organism at a given time. Taking an example used by the original proponents of the theory, the direct attentional process when looking at an apple would only provide the agent with information about the visual properties of the apple. In contrast, the attention schema would include not only information about the apple, but information about the "looking at the apple," and thus serve as a basis for more complex behavior as a result.

AST is supported by experimental evidence which suggests that a dissociation between awareness and attention is possible, with only the former corresponding to conscious experience, while the latter still contributes to behavior, even if that behavior is not consciously mediated (Kentridge et al., 1999). Such a dissociation has been found in other contexts as well, adding support to the theory (Koch & Tsuchiya, 2007). The temporoparietal junction in particular is implicated in the function of the AST (Webb et al., 2016). It is also likely that the prefrontal cortex is likewise involved in supporting the attentional schema, given its role in the representation of a variety of other high-level cognitive schemas (Gilboa & Marlatte, 2017). Furthermore, AST can be interpreted as being related to the "perceptual reality monitoring" hypothesis, whereby information only becomes consciously accessible when there is a match between high-level beliefs and incoming sensory information (Lau, 2019). This would correspond to an alignment between the content of the attentional process and the predictions of the attentional schema. In the perceptual reality monitoring theory, the prefrontal cortex plays a key role in this meta-cognitive process, providing additional evidence for the regions importance in AST as well.

### 3.4   Higher Order Thought Theory

There is an additional functional theory of consciousness which is often brought up when discussing those proposed here: Higher Order Thought Theory (HOT) (Gennaro, 1996). HOT proposes that it is only meta-represented information which is consciously accessible (Flavell, 1979), and these meta-representations are thought to be made possible via the activity of the prefrontal cortex (Lau & Rosenthal, 2011). While relevant to the topic of this work, we choose to not treat this theory separately for a number of reasons. The first is that from a functional perspective, all three of the theories described above propose that the contents of consciousness correspond in some way to a higher-order representation, with the GWT describing a multi-modal representational workspace, IGT describing a higher-order representation of sensory information, and AST describing a meta-representation of the attentional process itself. Because of this overlap, we believe that greater explanatory power can be derived from considering the separate mechanisms proposed by each of these three theories.

## 4   Defining intelligence

With an understanding of various approaches to formalizing conscious function, it is possible to turn to the second half of the link discussed above: intelligence. Already we have seen that various theories of conscious function imply specific cognitive abilities, but what is the link between cognition and intelligence, and how can intelligence be properly formulated? Despite its seemingly more formalizable nature, the history of intelligence as a concept has been almost as troubled as that of consciousness, with much work conducted over the decades both from the perspective of measuring human (Cattell, 1987), as well as machine intelligence (Legg & Hutter, 2007). The difficult nature of defining intelligence has largely arisen from the problem of dissociating specific anthropocentric abilities which humans possess from some more general principle which

would be agnostic to humans, and include not only other animals, but ultimately artificial agents as well. The refinement of such a definition involves the move away from measuring intelligence as the proficiency at executing specific individual skills such as language or tool use. Instead, a non-anthropocentric definition of intelligence can only exist at a more abstract level, referring in some way not to the ability to execute specific skills, but rather the ability to learn those skills, or indeed the ability to learn from a space of conceivable skills. In this definition, simply because an agent can execute a complex skill does not mean that that agent is particularly intelligent. Such a definition also accounts for the in-built skills provided via natural selection to various animals.

To properly define intelligence, more specificity is needed than simply to say that to be able to acquire skills is related to intelligence. This is apparent if we consider the fact that someone who has acquired the most skills is not necessarily the most intelligent, but perhaps may simply have had more time to do so. Instead, we can define intelligence as the efficiency at acquiring novel skills with little experience, little prior knowledge, and little structural disposition toward that skill. This is the definition put forward by Chollet (2019), following others. Such a definition can even be mathematically formalized as the speed of skill acquisition divided by the amount of experience, knowledge, and structural priors needed in order to acquire those skills. The specific quantification of these terms can be realized in different ways, with the authors utilizing a measure based on algorithmic information theory. Despite the choice of quantification, what is essential to this definition is that it is in principle quantifiable and universally applicable to any agent, whether natural or artificial. Furthermore, given this definition, we can say that to be more intelligent is also to be more generally intelligent, since greater intelligence implies the ability to apply one's intelligence to the acquisition of a broader range of potential skills. As such, "general intelligence" rather than being a binary property of an agent describes a scale of generality.

## 4.1 Intelligence in humans and other animals

With the above definition of intelligence, we can begin to make sense of claims concerning human and animal intelligence. Almost all animals display some form of learning, often in the form of basic associative learning displayed even by most insects (Hollis & Guillette, 2015). The range of possible skills which can be learned, and the amount of experience and prior knowledge required in order to learn these skills however varies wildly from species to species, and from individual to individual (Shettleworth, 1993). In principle however, the definition above provides a means of quantifying intelligence within these animals.

Given Chollet's definition of intelligence, what is of interest in humans is their ability to learn to perform novel skills with little to no prior experience or knowledge of those skills. This is in contrast to many animals which can be trained to learn novel conceptual skills, but require orders of magnitude more experience to do so than humans (Zentall et al., 2008). At the extreme end is the ability to perform what is known as few-shot or zero-shot learning in the machine learning literature, which can apply straightforwardly to animal cognition as well. In the case of few-shot learning, only a small number of trials are required to learn to perform a novel skill. In the case of zero-shot learning, the skill can be performed on the first trial, by successfully generalizing from related experience.

We can understand the capacity for rapid learning more acutely in the domain of language acquisition. While humans benefit from specific evolutionary advantages regarding vocally expressed language, both humans and chimpanzees are physiologically capable of deploying sign language using their limbs and hands. Despite this, even when chimpanzees have been taught to use symbolic communication in the form of sign language, an extremely impressive skill, they do so in a limited and stereotyped manner (Gardner & Gardner, 1969; Fouts, 1972). They likewise learn new words at a rate significantly slower than that of a human child. In cases where chimpanzees have generated sign language in the absence of experimenter prompting, it is often for the expression of bodily needs or desires such as specific kinds of food or play, rather than in a manner which supports the rapid transmission of novel skills between animals.

The example of language learning raises an important consideration regarding the definition of intelligence put forward by Chollet. While humans are thought to display greater domain-general intelligence compared to most other animals, they do benefit from various structural priors which condition the domains in which they can display that intelligence. Language has been hypothesized to be one such domain, starting with the

work of Chomsky et al. (1976) who suggested that humans possess a structural prior for language acquisition. While the exact role and scope of this prior is still debated, it is thought to at least provide some meaningful disposition toward language learning in infants (Kuhl, 2000). The difficulty then becomes weighting the relative importance of structural priors compared to previous experience and knowledge when comparing the intelligence of two or more agents. If one believes that humans possess a strong prior for language, then they may be less impressed by humans displaying orders-of-magnitude faster language acquisition compared to chimpanzees.

It is unreasonable to expect that an agent would display more rapid skill acquisition than another with no structural priors at all however. Therefore, in order to practically deploy Chollet's measure of intelligence, the question becomes which priors are both relevant to the kinds of skills we might be interested in, and are general enough to be applicable to a wide array of skills we may not be able to anticipate. A group of theorists of human intelligence including most prominently Spelke & Kinzler (2007) have proposed a set of four structural priors which are referred to as the "Core Knowledge" which they propose that all humans possess to some extent. These correspond to the ability to represent objects, actions, numbers, and space. In agreement with Chollet (2019), we believe that these four priors represent a reasonable restriction of the domains of intelligence (and thus plausible tasks) which AI researchers might be interested in. As such, one of the key tasks of artificial intelligence becomes how best to instantiate these priors in a general purpose manner which enables the reduction of required experience or knowledge for any given skill acquisition.

## 4.2 Intelligence in artificial agents

The past decade has seen rapid progress made in the field of AI. This is evidenced in not only a large breadth of published research, but also the development of systems capable of solving what were thought to be long-standing challenges within the field, such as recognizing objects within images (Krizhevsky et al., 2012) or playing games such as Go or poker at a super-human level (Silver et al., 2016; Moravčík et al., 2017). This has largely been driven by one particularly successful method within AI: deep learning, which utilizes deep artificial neural networks to model data (LeCun et al., 2015; Schmidhuber, 2015). These achievements have led some to propose that artificial general intelligence (understood to be an artificial agent with intelligence of the kind defined here equal-to or greater-than that of humans), must surely be on the near horizon, with only larger models trained on even more data needed to one day achieve it (Sejnowski, 2020). Others have met these successes with relative skepticism (Marcus, 2018). The criticism being that artificial agents, while seemingly intelligent in their narrow domain of expertise, often fail to generalize to even small changes in the task distribution.

Examining the agents behind these successes more closely, we find that they do indeed often fail to live up to the definition of intelligence provided by Chollet (2019). In order to arrive at an agent capable of high-accuracy image classification, or grandmaster-level Go playing, millions to billions of training examples are required to be processed and learned from (Dai et al., 2021; Silver et al., 2016). Processed serially, such training data is equivalent to decades or even centuries of human experience, far less than what humans actually require to perform these tasks (Lake et al., 2017). This alone would suggest that these agents are relatively unintelligent by human standards. Further compounding this issue is the fact that these systems are the products of dozens to hundreds of highly skilled engineers and scientists working together to produce neural network architectures fitted for the specific problem domain, as evidenced by two recent high-profile works in the domain of real time strategy game-playing involving dozens of authors and many more acknowledgements each (Berner et al., 2019; Vinyals et al., 2019). This amounts to a tremendous amount of domain knowledge and structural priors embedded into any such model even before the formal learning process begins. Because of this, these agents are ultimately limited in their application to the domain of experience and built-in prior knowledge with which the system was provided.

Within the field, there have generally been two approaches to dealing with the apparent lack of intelligence in so much of AI. On the one hand, there is the perspective that models which are orders of magnitude larger, and trained on orders of magnitude more data, will be able to display properties of few-shot generalization like those found in humans (Schmidhuber, 2018). Indeed, some experimental evidence even supports this possibility, with the double-descent phenomenon suggesting that orders-of-magnitude-larger models are able to overcome the over-fitting problems of typical large neural networks (Nakkiran et al., 2019). The capacity

for much larger models to serve as a basis for generally intelligent agents has been the promise of recent so-called foundation models (Bommasani et al., 2021)—such as GPT-3 (Brown et al., 2020)—with the latter demonstrating zero-shot adaptation to language-based tasks, without explicitly being trained to do so. This approach has also been extended to artificial agents, with the procedural generation of environments and tasks leading to zero-shot generalization and more adaptive behaviors within a multiagent setup (Team et al., 2021).

The more traditional approach to addressing the lack of intelligence in AI has been to attempt to develop novel methods which enable greater generalization in a more principled fashion. Indeed, the problem of generalization, and thus intelligence as defined above is one of the major active areas of research within the field. In the following section, we provide a deeper treatment of some of these approaches.

## 5   Approaches to generalization in artificial intelligence

We center our discussion of approaches to AI generalization around machine learning, in which a model is optimized on a set of data in order to perform a task. At a fundamental level, how well the model generalizes depends on the optimization process, the dataset, and the model itself. While many settings within machine learning utilize static datasets, of particular relevance to our discussion are "agents" that can interact with their environment, being embodied in some form (Shapiro, 2010). Indeed, some have proposed that it may be impossible for the aforementioned large models, trained purely on language, to understand things the way that humans do without being grounded in the real world (Bender & Koller, 2020; Bisk et al., 2020).

### 5.1   Reinforcement learning

One way of formalizing decision-making agents is provided by reinforcement learning (RL) (Sutton & Barto, 2018). The basic RL setup is thus: at any moment in time, the agent observes the current state of its environment (including any state internal to the agent itself), and can perform an action in response in order to change the state. Every change in state is accompanied by a (scalar) reward signal, and the objective of the agent is to adapt its policy—its selection of actions conditional on the state—in order to maximize its return (cumulative reward) over its lifetime. Ecologically-relevant extensions (Vamplew et al., 2021) include goal-conditioned RL (Liu et al., 2022) and multi-objective RL (Roijers et al., 2013), which allow the agent to consider utility functions that vary in a task-dependent fashion. States, actions, rewards—and potentially goals—comprise the agent's data, and its policy either explicitly or implicitly captures knowledge of the world around it. Given RL's focus on temporally-extended decision making under uncertainty, it is used as a basis for at least one formalization of general intelligence (Hutter, 2004).

The considerations of interactivity and (future) return separates RL from other paradigms within machine learning. In order to accomplish a complex task, an agent needs to be aware of the affordances of the environment, and be able to perform long-term credit assignment in order to associate past actions with future rewards. A common way to help accomplish this is to learn a value function, which estimates the return available from a given state (and given action, depending on the form of the function). Another common consideration to improve the data-efficiency of RL algorithms is to incorporate off-policy learning: whilst an on-policy algorithm is limited to its current experience, an off-policy algorithm can update itself based on trajectories generated from other behavioral policies than the current target policy—where the former includes past experiences of the current agent.

### 5.2   Model-based reinforcement learning

A basic premise of RL is that the agent is not given an explicit model of the world, and is therefore restricted to learning about its environment through interaction. As such, the agent can form a policy by explicitly learning and using a model of the environment (model-based RL; deliberative behavior), or without doing so (model-free RL; habitual behavior). From the RL perspective, an environment model consists primarily of the transition dynamics (how, given the current state, the agent's actions will determine the next state), but typically also includes a prediction over the next reward as well.

While model-free RL, incorporating policies and/or value functions, is a core component of decision making in biological agents (Niv, 2009), and has even achieved superhuman results in virtual domains (Berner et al., 2019; Vinyals et al., 2019; Badia et al., 2020), we tend to relate model-based RL more closely to the concept of intelligence in animals (Hamrick, 2019). The benefits are clear—with a model, one can simulate the consequences of policies without having to interact with the real environment, or even explicitly plan or reason about goals. This approach has enabled artificial agents to learn policies which are competitive with model-free counterparts (Hafner et al., 2019), but utilize orders of magnitude fewer samples, thus being strictly more intelligent in the sense presented here.

We note that this does not preclude the importance of model-free components in intelligent agents. For instance, model-free temporal-difference learning allows (effective but aliased) credit assignment over long time horizons (Sutton, 1988), and is well-studied in animals (Schultz et al., 1997). There also exist possible hybrids between model-free and model-based approaches (Dayan, 1993), with evidence for such a strategy in humans (Momennejad et al., 2017).

## 5.3   Inductive biases for generalization

More broadly, given a finite number of observations about the world, can one make successful inferences over novel data? This of course depends on what assumptions are built into one's model of the data. A broad assumption is the minimum description length principle (Grünwald, 2007)—a formalization of Occam's razor—which states that the simplest explanation of the observed data is most likely to be the true explanation. AI researchers can also build inductive biases within their models, a technique often used within deep learning (Bengio et al., 2013). For instance, we know that (images of) natural scenes contain many local spatial correlations (Field, 1987), and convolutional neural networks, which have translation equivariance built in (Bronstein et al., 2017), successfully exploit this property, making them the most popular form of AI model used within computer vision applications (Khan et al., 2018).

One general inductive bias is that of sparsity (Hastie et al., 2015). Given a high-dimensional observation of a real-world phenomena, only a subset of that observation is typically relevant at any one point. Beyond purely static variable selection at the input level (George, 2000), only a subset of high-dimensional signals may be relevant in a given timeframe. This leads to the idea of applying attention over both external and internal signals. And as the state of the environment changes over time, so does the subset of information that needs to be attended to. Why not process all information at once? One could argue that any real agent would have bounded compute resources (Simon, 1990). However, there is also a more fundamental reason when it comes to modelling the world accurately—reasoning over too many variables may lead to learning spurious correlations (Simon, 1954). Spurious correlations relate to causal relationships, which we discuss in Subsection 5.6.

A related inductive bias is that of modularity. While much has been made of the homogeneity of neuron structures such as cortical columns (Hawkins et al., 2019), at a higher level we subdivide the brain into regions, such as the amygdala or prefrontal cortex—or even divide the visual cortex into several levels. While it is conspicuous to deal with different sensory modalities via specializing processing, applying a divide-and-conquer paradigm is a common strategy more broadly within AI (Jacobs et al., 1991; Masoudnia & Ebrahimpour, 2014). Within deep learning, modular neural networks have been popular in the multitask setting (Rusu et al., 2016; Fernando et al., 2017; Rosenbaum et al., 2017), extending to the use of composable modules that can be rearranged and reused in more arbitrary ways (Alet et al., 2018). Similarly, modular networks are a common solution under continual learning settings (Parisi et al., 2019). From a computational perspective, the advantages of modularity are that it allows for a separation of concerns, reduces interference, and can make inference more tractable by only requiring the use of a subset of the model.

## 5.4   Attention and adaptive processing

As a way of performing inference selectively over inputs, attention in artificial neural networks has been studied for many decades (Schmidhuber & Huber, 1991), but has only relatively recently become a major component of mainstream machine learning models with the advent of the Transformer architecture (Vaswani et al., 2017). Like early work in neurosymbolic architectures (Smolensky, 1990), Transformers use the

agreement between keys (associated with values) and queries in order to attend to specific parts of the input. This enables the processing within the network to become context-dependent—elements within the input are processed differently based on the other elements within the input. Transformers can also be linked to earlier work on "fast weights" (Schlag et al., 2021), where a portion of the "synaptic weights" within the network adapt online to current inputs, without requiring further training (Hinton & Plaut, 1987; Schmidhuber, 1992). Attention, as well as other forms of adaptive computation (Graves, 2016), can be thought of as a way of focusing computation on where it matters, both at the input level, and within the model itself.

### 5.5  Abstraction and multimodality

Although neural networks (both artificial and biological) might have distributed representations, phenomenologically we are able to reason about discrete objects and their relationships, suggesting that we are able to transform low-level sensory inputs into abstract concepts. While there are arguments for explicitly building symbolic reasoning capabilities into neural networks (Marcus, 2020), we will discuss what "standard" neural networks are capable of already.

The fundamental principle behind deep learning is hierarchy in representations (Bengio et al., 2013). Driven by data, deep neural networks given image pixels as input will find edge and texture detectors (Zeiler & Fergus, 2014), and similarly derive phonemes and words from speech (Lee et al., 2009). Within a single neural network, these learned representations will become more abstract the closer they get to the output of the network. Despite the debate over the importance of neurosymbolic approaches (Marcus, 2020; Garcez & Lamb, 2020), for the time being it seems that simply scaling architectures and data results in models that are able to reason over progressively higher-level concepts (Brown et al., 2020; Radford et al., 2021).

A more precise route to further abstraction and hence intelligent behavior is the ability to incorporate multimodal information. From a machine learning perspective, multimodality can be considered a regularizer—any representation derived from more than one modality should be consistent across all modalities, and hence more robust to spurious associations that might exist in only one modality (Srivastava & Salakhutdinov, 2012). For example, if one were to consider the concept of a dog, it extends beyond visual appearance to the sounds it makes, the feel of its fur, its smell, and even its typical behaviors; a single one of these would be insufficient to create the concept. In neuroscience, the study of Quiroga et al. (2005) famously found a "Jennifer Aniston" neuron in one subject—a neuron that singly responded not just to images of the famous actress, but also text with the name. Recent work investigating a large-scale text-and-vision model Radford et al. (2021) has found a similar singular neuron for Spider-man—responding to both images and the name of the superhero (Goh et al., 2021).

### 5.6  Causality

Although an environment model, which is sparse and modular, with high-level objects, is potentially very powerful, if it is purely correlational then it can fail to make the correct inferences under distribution shift. Something more is needed in order to deal with a world that is changing and partially-observed.

To make this more concrete, we consider the notion of "out-of-distribution" (OoD) generalization (Salehi et al., 2021). A typical assumption within machine learning is that the distribution of the test data matches that of the data used to train a model. However, even if we've never seen a camel outside of a desert, a human would be able to recognize the same camel as such even in a green pasture—unlike typical AI systems (Arjovsky et al., 2019). Using the field of causality (Pearl, 2009), one definition of OoD is that a model must be robust to "interventions" in the data-generating distribution (Arjovsky et al., 2019). In causal learning, one assumes the world consists of objects that have causal relationships with other objects, and in practice, most objects are independent of each other. While machine learning broadly considers associational models, i.e., modelling correlations between variables, causal models aim to capture the underlying cause-and-effect relationships. For example, the presence of umbrellas indicates rain, as rain causes people to put up umbrellas, but the intervention of deciding to put up an umbrella does not cause rain to fall. Therefore, beyond evidence that humans learn causal models (Waldmann & Hagmayer, 2013), we posit that causal models are needed for sufficient generalization.

The true power of causal models lies in their ability to enable counterfactual reasoning. While interventions, exemplified by randomized control trials in medicine, are a powerful tool for causal discovery, they may not be possible to carry out in reality. Counterfactual reasoning instead takes an existing observation, and performs the following simulation: if one variable were different but everything else were the same, what would the outcome be? Agents with counterfactual reasoning can therefore make some novel inferences without having to perform further actions (i.e., interventions) in the world.

### 5.7 Meta-learning

A final area of research relevant to general AI is the ability to not just learn a new task, but to "learn to learn"—also known in the literature as meta-learning (Schmidhuber, 1987). Given a distribution over tasks, an agent that can meta-learn should be able to learn new tasks more easily over time: one could learn how to play badminton quicker after having learned tennis, and learn squash even quicker having learned two racket sports prior. Chollet (2019) specifically links meta-learning to his definition of intelligence, as it pertains to not just skill acquisition, but rate of skill acquisition.

In the AI literature, there are three main approaches to meta-learning: metric-based, optimization-based, and memory/model-based (Weng, 2018). Metric-based methods can be related to episodic memory (Ritter et al., 2018)—given a new datapoint, the model's interpretation of a novel datapoint is based on a weighted average of previously observed datapoints in memory. Given that learning is typically formulated as adapting (explicit or implicit) parameters, optimization-based methods train a model such that its parameters can be efficiently updated by an optimizer on just a few novel datapoints. Finally, model-based methods contain a subset of parameters—typically considered the model's "memory"—that quickly update in the presence of new data. The latter, implemented using recurrent neural networks, are not only one of the simplest ways to perform meta-learning in an RL setting (Wang et al., 2016; Duan et al., 2016), but are also near-optimal in the Bayesian sense (Ortega et al., 2019). While each approach has its advantages and disadvantages, at a minimum, agents with some form of memory are capable of rapid adaptation, and hence are pertinent to our discussion of intelligence.

## 6 Mental time travel: a motivating example

With an understanding of conscious function and intelligence in humans and machines developed, it is now possible to make the link between the two explicit. We do so with a motivating example. According to the functional theories proposed above, access consciousness is enabled by the selective maintenance and manipulation of subsets of information generated through internal simulation, and does so in a way which is conditioned on a meta-representation of the attentional process itself. An explicitly conscious cognitive ability which requires all of these is mental time travel (Tulving, 2002). Mental time travel is also relatively unique because it is one of the few cognitive abilities for which there exists compelling evidence of it only being observed in humans (Suddendorf et al., 2011) (though for conflicting evidence, see (Roberts, 2007)). Of particular relevance to our overall discussion is its relationship to intelligence: mental time travel is driven by top-down processes that enable us to reason about situations beyond our current environment and needs (Clayton et al., 2003). Anticipatory planning, a special case of mental time travel, is conjectured to have been present in prehistoric hominins, allowing for tool construction and other displays of intelligence beyond non-human primates (Osvath & Gärdenfors, 2005).

The phenomenon of mental time travel was first formalized by Tulving in the mid 20[th] century as a description of a specifically human ability associated with episodic memory (Tulving, 1985; 2002). Tulving describes mental time travel as the ability to project oneself into a past or future experience. Mental time travel is a first-person experience, going beyond pure knowledge retrieval from semantic or episodic-like/what-where-when memory (Clayton et al., 2007; Suddendorf & Busby, 2003).[1] In humans, it is tied to causal reasoning, as one does not merely re-experience that event from the perspective of a passive observer, but can intervene on the event in novel ways as if one was actually there. Importantly, the individual performing mental time

---

[1]Though of course retrieval exists along a spectrum, with semantic memory compensating for a lack of episodic detail during mental time travel (Devitt et al., 2017).

travel is both aware of the simulated experience and also aware of the fact that it is a simulated experience. According to Tulving, this ability of humans is made possible through what he referred to as "autonoetic consciousness," or a sense of subjective lived time. These subjective, projective, and prospective aspects of mental time travel make them distinct from the mere ability to simply recall semantic information or static episodic events stored in memory.

We can unpack the phenomenon of mental time travel with respect to the cognitive sub-components which contribute to make it possible. It is first clear that in order to re-experience past experiences or imagine future experiences, one needs a system by which chronological (but not necessarily temporally contiguous) trajectories can be simulated. The IGT provides a description of such a mechanism within the brain. During the process, one must suppress the current (lived) experience, and, at any given time, one is only aware of specific relevant aspects of the imagined experience, which are attended to and drive the subsequent trajectory of the experience. The GWT provides a description of such a system of attention, maintenance, and broadcasting. Finally, in mental time travel one is aware of oneself as a "time traveler," and is thus able to adapt one's attentional policy over the course of the experience from a first-person perspective. Here we find that AST provides a compelling description of this capacity to model attention in the form of meta-cognitive awareness—including self-awareness. Removing any one of these components makes mental time travel in its full definition no longer possible.

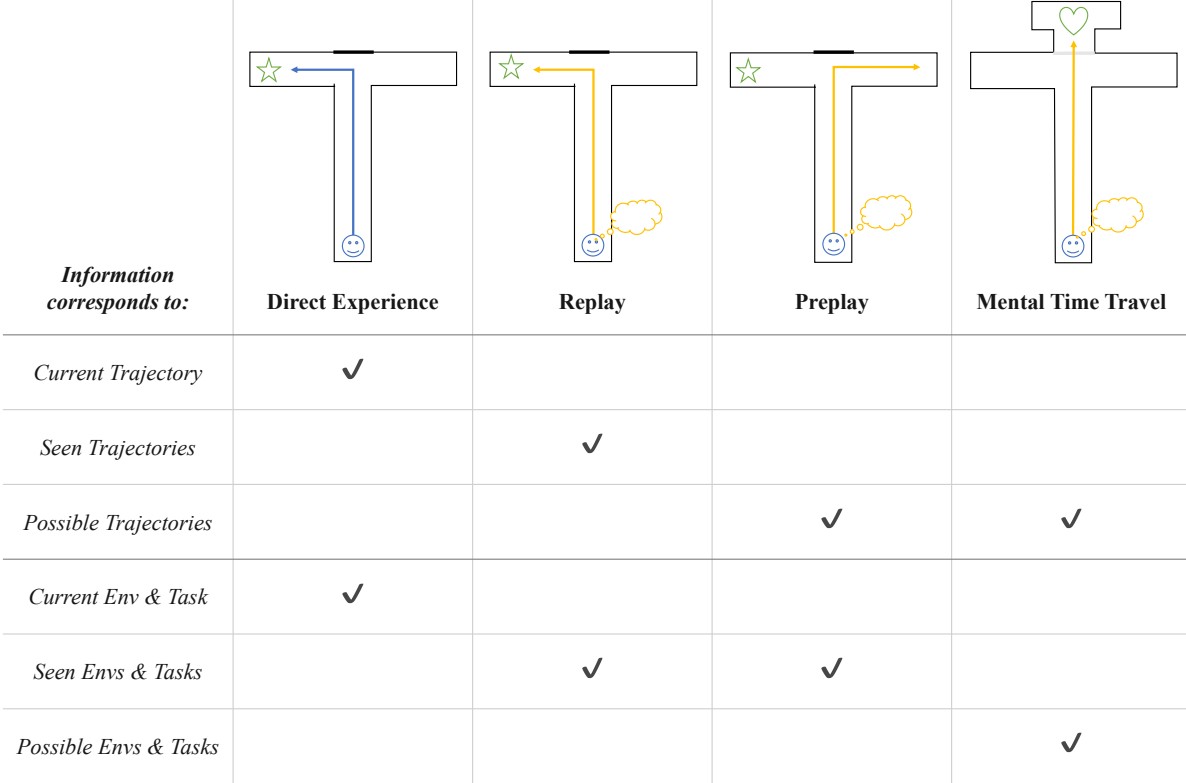

| Information corresponds to: | Direct Experience | Replay | Preplay | Mental Time Travel |
|---|---|---|---|---|
| *Current Trajectory* | ✔ | | | |
| *Seen Trajectories* | | ✔ | | |
| *Possible Trajectories* | | | ✔ | ✔ |
| *Current Env & Task* | ✔ | | | |
| *Seen Envs & Tasks* | | ✔ | ✔ | |
| *Possible Envs & Tasks* | | | | ✔ |

Figure 2: Schematic of the four levels of experience generation. **Left**: Direct Experience. A trajectory of observations and actions through the current environment. **Middle-Left**: Replay. A simulation of already experienced observations and actions. **Middle-Right**: Preplay. A simulation of possible observations and actions given the current environment and behavioral needs. **Right**: Mental time travel. A simulation of possible observations and actions drawn from environments and policies, given current and potential future behavioral needs.

### 6.1 Levels of experience generation

The question then arises of how exactly mental time travel in humans may differ from other forms of memory, imagination, or simulation which other animals and artificial agents are readily capable of. The importance of this distention is particularly important in light of the frequent use of the term "mental time travel" in more broad contexts than originally proposed by Tulving, such as to describe what takes place during vicarious trial-and-error in rodents (Tolman, 1939; Redish, 2016). In order to better clarify mental time travel and contrast it with other related cognitive phenomena, we outline a hierarchy of four levels of experience generation, with proper mental time travel being the top level, and thus the most complex. Note that while the levels are presented as discrete here for explanatory purposes, this hierarchy can be best thought of as a continuum, with many animals or artificial agents displaying abilities which could be seen as in-between various levels. For a schematic representation of these different levels of experience generation, see Figures 2.

**Direct Experience**: The first level we consider is that of direct experience, which consists of a trajectory of observations and actions taken by an agent. At this level, the agent can learn only from the current trajectory of experience itself, and the opportunity for generalization is limited, being bounded by the spatial and temporal constraints of the agent.

**Replay**: The second level we consider is experience replay, in which an agent can generate previously seen trajectories of observations and actions. This is done in order to incorporate useful information from those experiences into the current skill learning process. In order for replay to be possible the agent can store previous experiences as-is, but more sophisticated forms of replay can also involve learning a generative model of the manifold of observations and actions the agent has experienced in the past. Such a model can then be sampled from to generate trajectories without the need to store all previous experiences.

**Preplay**: The third level of the hierarchy is experience preplay, whereby an agent is able to simulate not only previously seen observations and actions, but also possible future observations and actions. This corresponds to what is often referred to as vicarious trial-and-error in the literature (Tolman, 1939). The flexibility of preplay is however limited to generating observations from seen environments and tasks in those environments, and is limited to the current behavioral context of the agent. This requires learning a manifold of both the current environment the agent is situated within, as well as the relevant policies for seen tasks, such that the simulation can sample from these manifolds to generate imagined trajectories.

**Mental Time Travel**: The fourth and final level in our hierarchy of experience generation is mental time travel. At this level, the agent is able to generate trajectories from not only visited environments and policies induced by seen tasks, but unseen possible environments and policies induced by possible tasks, thus enabling a much larger distribution of generated experiences to learn from. This requires learning the manifold of what we refer to as the current world (collection of possible environments) the agent find itself within, as well as the manifold of the agency (possible policies from possible tasks). The space of possible tasks in mental time travel is often based not only on current behavioral needs, but also on anticipated future needs. Animals and artificial agents are capable of various levels of this hierarchy, though we are only aware of compelling evidence for humans meeting the criteria proposed for mental time travel with respect to environments sampled from the physical world (Suddendorf et al., 2011).

### 6.2 Experience generation in humans and other animals

We can first compare humans and other animals to understand at what level of the hierarchy they may be categorized. In mammals such as rodents there exists a phenomenon referred to as hippocampal replay. A replay event corresponds to trajectories of place cells and their attendant neighboring cells such as grid and time cells reactivating in sequences which are consistent with those of previous waking experience (Davidson et al., 2009). Place cells each correspond to a specific location in space-time that an animal can find itself within. These spontaneously generated trajectories can occur when the animal is at rest, sleeping, or about to make a decision, thus making up a large portion of the activity of the hippocampus. Replay events have been linked to a variety of different cognitive functions, including memory consolidation, policy distillation,

and planning (Foster, 2017). Because of this, it is tempting to draw a correlation between mental time travel and replay events in mammals, as some researchers have done in the past (Hasselmo, 2009).

The phenomenon of replay can however be distinguished from mental time travel in a number of important ways. In contrast to mental time travel, which proceeds in a forward order and is experienced as subjectively lived, replay events occur at orders of magnitude faster than the original experience, and can take place in reverse order (Foster & Wilson, 2006), or be interwoven between multiple unrelated trajectories (Kay et al., 2020). Contrasting them further, replay events correspond to already experienced state and action trajectories rather than unseen environments or novel policies.

A related body of literature describes the phenomenon of preplay in mammals. In preplay, the place cell trajectories in the hippocampus correspond to not past experience, but rather to the future experience of the animal (Ólafsdóttir et al., 2015). Preplay can thus be thought of as a form of prospective experience generation, whereby an animal may plan out or learn from potential future behavior (Pezzulo et al., 2014). This projective behavior has been associated with vicarious trial-and-error behavior in rodents, where animals will pause at decision points and move their attention between action choices (Redish, 2016). The term "preplay" has also been used in the literature to refer to the fact that the hippocampus recruits template-like trajectories of hippocampal activity to encode novel experiences Dragoi & Tonegawa (2011). While interpretation of this phenomenon has been debated (see (Foster, 2017) for a review), we use it here in the more general sense of behaviorally relevant experience generation, following Pezzulo et al. (2014) and others.

Once again however the role of preplay is meaningfully different from that of mental time travel. Whereas preplay is limited to the current environmental context and behavioral needs of the animal, mental time travel involves possible environments and possible policies, a much broader class. Furthermore, the extent of generalization possible during animal preplay has been a topic of debate, and may be relatively limited (Foster, 2017). Like replay, preplay events also occur in rodents at an order of magnitude faster than physical experience, and are thus unlikely to be consciously experienced by the animal.

Given the continuous nature of the levels of experience generation, we would expect that animals with a close evolutionary connection to humans such as great apes to display similar cognitive abilities, and thus be placed on a similar place in our experience generation hierarchy. While there has been considerable debate as to whether chimpanzees or other apes are capable of simulating experience in the service of future behavioral needs (Suddendorf & Corballis, 2008), recent experimental work suggests that they are at least capable of a sophisticated form of preplay, if not even of rudimentary mental time travel. In a series of experiments, Osvath & Osvath (2008) demonstrated that great apes were able to select a tool required for the acquisition of a desirable reward up to an hour before the availability of said reward. Furthermore, they selected the tool over a potential current reward of lesser value than the tool-conditioned delayed reward. Importantly, after using the initial tool they were able to generalize to an unseen tool which served the same purpose, and select it when available as well. This work suggests the ability for apes to suppress current behavior needs in the service of known future behavioral needs at relatively long timescales. The ability to select a novel tool for the same purpose also suggests some capacity for future-oriented planning. The study does not however demonstrate zero-shot learning, since the apes were all exposed to the tool-reward contingencies by the experimenter beforehand. In the absence of the ability to probe the mental content of the apes, it is also impossible to determine whether mental time travel actually took place, as opposed to a complex semantic association between the tool and the delayed reward.

### 6.3 Experience generation in artificial agents

Current research in AI, and particularly RL, covers a spectrum of experience generation capabilities. The most limited are on-policy RL algorithms, which are only able to learn from direct experience. Whilst they can use the temporal structure of trajectories to perform credit assignment, their data-efficiency is fundamentally limited, only outperforming black-box optimization algorithms.

When limited to learning from past experience, off-policy RL algorithms correspond to the second level of experience generation. In fact, the seminal work in deep RL (Mnih et al., 2015) popularized the hippocampal-replay-inspired "experience replay" (Lin, 1992), in which trajectories are stored and replayed from a large circular buffer for learning from. These trajectories can be selected at random, or prioritized based on metrics

of behavioral relevance (Schaul et al., 2015; Isele & Cosgun, 2018). This approach provides the policy improvement and memory consolidation benefits of hippocampal replay, but has only extremely limited generalization beyond observed states and actions: technically, state-action value functions enable one-step interventions with novel actions (Mesnard et al., 2021).

Whilst the majority of deep RL algorithms use experience to update the policy and/or value functions offline, some RL methods have taken direct inspiration from episodic memory to create policies that are directly conditioned on past experiences (Blundell et al., 2016; Pritzel et al., 2017). Sufficiently large buffers of experiences, which can be queried as needed during the learning of a novel task, provide powerful retrospective learning (Lampinen et al., 2021), but limited future-oriented imagination.

Current methods in MBRL can largely be mapped to replay, with some more sophisticated methods being capable of preplay as well. A common form of these algorithms is a decomposition of the world model into an observation model—mapping observations to a latent state space—and a transition model over the latent state space (Ha & Schmidhuber, 2018; Hafner et al., 2019; Schrittwieser et al., 2020). When trained on observations from an agent, these methods construct a generative model of the experienced environment which can be used for investigating the impact of novel states and actions. Whilst such models could be queried with the agent's current policy, in practice search algorithms like the cross-entropy method (Rubinstein, 1997; Hafner et al., 2019) or Monte Carlo tree search (MCTS) (Coulom, 2006; Schrittwieser et al., 2020) enable learning from unseen, task-relevant trajectories. In order for preplay-like learning to be successful, learned models must generalize beyond observed states and actions. This can be achieved through either increasing the training data, or incorporating relevant inductive biases: for instance, using geometry for localization within 3D Euclidean domains (Rosenbaum et al., 2018; Fraccaro et al., 2018).

There has been limited progress in extending MBRL methods further. Meta-learning or multitask training for models within MBRL enables rapid adaptation to novel tasks/dynamics within an environment (Nagabandi et al., 2018; Landolfi et al., 2019), and learning structural causal models allow policy updates to be explicitly grounded on observed trajectories via counterfactual reasoning (Buesing et al., 2018), but these have only been demonstrated in domains with relatively simple dynamics. However, beyond limited generalization, current methods still lack several key components of mental time travel. In particular, a top-down, proactive process has to prioritize the use of the agent's world model to generate experiences—which are not necessarily in service of the agent's immediate needs—and simultaneously suppress its current experience. Within the generation, the agent exists as a distinct entity to its environment and is able to perform interventions that go beyond its observed experiences—or even what is possible in the environment.

### 6.4 Mental time travel in the "real world"

One important point to make is that the different levels of experience generation require different computational implementations depending on the complexity of the underlying state and action spaces of the domain of environments under consideration. In very simple environment domains such as 2D grids which obey straightforward Euclidean geometry, it may be possible to perform a form of mental time travel by learning a generative model of the manifold of possible grids. We would expect that such an agent would be more intelligent than similar agents with only replay or preplay abilities within that domain. That said, we would expect an agent capable of mental time travel in a more complex domain, such as the space of possible Atari 2600 games to be significantly more intelligent than the mental time travel agent in the gridworld domain. Extending this principle further, humans evolved to perform mental time travel in the domain of the physical or "real" world, where the complexity and variety of environments varies greatly. As such, we expect agents such as humans who are able to perform mental time travel in this domain to be significantly more intelligent than agents in other less complex domains, and this is indeed the case. Likewise, we expect more complex mechanisms to be required in order to enable mental time travel within the "real world" compared to virtual abstract domains.

One thing which all three of the domains discussed above all share are a state space which assumes a certain amount of spatial and temporal coherence. In these worlds, time typically proceeds in a forward fashion, and any two states are near each other in space, whether that space be 2D or 3D. The ability to learn models of these domains which are capable of replay, preplay, and ultimately mental time travel rely on the

domains displaying these consistent structural properties from which generalization is then possible. These domains need not be strictly physical. In humans, there is evidence that the same cognitive processes which support mental time travel in the physical world are extended for use in other abstract domains such as social relationships where there is similar spatial or temporal structural coherence (Behrens et al., 2018). We can however imagine domains which do not possess these shared properties, and which may be significantly less amenable to human-like experience generation, such as the domain of all possible computer programs. This understanding of mental time travel is consistent with the set of "Core Knowledge" structural priors proposed by Spelke & Kinzler (2007) and endorsed by Chollet (2019), with the ability to reason about objects, agents, and space-time all being supposedly innate abilities in humans.

Given its inherently conscious nature, we can see mental time travel in the "real world" as a kind of Turing Test for access consciousness in artificial agents. While its absence does not suggest that the agent is not conscious (certainly not in the phenomenal sense), its presence serves as strong evidence for what many see as the markers of access consciousness found in humans. As such, there exists an asymmetrical relationship between the two concepts. Given our understanding of access consciousness and mental time travel, it also suggests that agents with the ability to perform mental time travel in real-world environments likely possess the main benefits of access consciousness: a more general intelligence. The benefits to intelligence of mental time travel are readily apparent. The ability to selectively simulate relevant aspects of either past or future experience enables one to acquire novel skills without the need to explicitly obtain experience or knowledge from the true environment. Furthermore, the domains in which mental time travel can be applied are not limited to specific kinds of skills. Humans are able to imagine themselves in unseen environments performing unseen tasks, such as flying in a rocket to the moon, and in doing so make such behavior a reality in a zero-shot manner.

## 7 Access consciousness and general intelligence

What makes it possible for an agent to perform mental time travel? We can ask this both about the humans who possess the ability, and the artificial agents we might want to possess such abilities in the future. We find that while not sufficient on their own, the systems described by the three functional theories of consciousness discussed here all have a key and necessary role to play in mental time travel. Likewise, the implementation and interplay of such systems have the potential to enable similar abilities in artificial agents.

### 7.1 A unified approach to access consciousness

To understand how the three functional theories of consciousness discussed here work together, we can use the classic example of mental time travel described in Marcel Proust's novel "In Search of Lost Time" (Proust, 2003). A scene early in the work describes how the taste of a madeleine cake dipped in tea brings back a flood of seemingly forgotten memories of the narrator's childhood. This stream of memories is then intensely felt as being re-lived by the narrator, and subsequently recounted for much of the novel. Proust's experience can be seen as relying first on the presence of a global workspace, where salient information in one sensory modality (smell/taste) enters conscious awareness, triggers an ignition event, and results in the retrieval of the lost memory. Rather than simply being semantically recalled however, the narrator finds that he is mentally re-living the experience of his childhood, thanks to the capacity to internally generate sequences of coherent experience which he possesses. Finally, he is acutely aware of the experience being a memory, and of himself being the one to re-experience thanks to the functioning attention schema which governs his awareness.

As made apparent in the example above, using mental time travel we can derive a unified perspective on how the three theories of consciousness presented here might work together to form a single coherent system. In this system, sensory and motor information is mediated by a process of internal experience generation. This system is itself spatially and temporally structured by the presence of a cognitive map. From a biological perspective, the recurrent top-down projections from higher to lower sensory areas may make internal experience generation possible (Clark, 2012; Manita et al., 2015). Likewise, the medial temporal lobe, in addition to playing a central role in episodic memory (Burgess et al., 2002), provides the computational architecture to maintain coherent spatial and temporal structuring through its connection to both downstream sensory

regions as well as upstream, more abstract cognitive regions (Hassabis & Maguire, 2009). Higher cognitive regions such as the orbitofrontal cortex may also contribute to the cognitive map (Wilson et al., 2014), which has been proposed to exist at multiple scales of abstraction at different regions in the brain (Brunec & Momennejad, 2022). These two systems of experience generation and experience structuring interact during the generation of experience, including, for example, when dreaming (Ji & Wilson, 2007).

The internally generated information is then available for selection, maintenance, manipulation, and broadcasting by a global workspace which operates an attentional policy over the space of generated information. This attentional process is over individual dissociable components of information, such as the redness of an object, or a particular feeling in the body (Mashour et al., 2020), as opposed to the entire sensory input of different modalities such as vision or sound. We believe this process of global maintenance and manipulation to be mediated largely by the fronto-parietal network, which takes input from both sensory and motor cortical regions, as well as more medial brain structures.

The dynamics of this attentional policy instantiated by the global workspace, over both inputs and outputs, is then modeled at a higher level by an attentional schema, which enables the adaptation of those dynamics based on the current and future behavioral needs of the system. Following others, we locate the attentional schema in the temporoparietal junction and its interaction with the fronto-parietal network (Graziano & Webb, 2015). In this way, we can see IGT, GWT, and AST forming complementary systems which each contribute a unique element to flexible information processing, and thus a form of relatively general intelligence. See Figure 3 for a diagrammatic representation of how these components may work together.

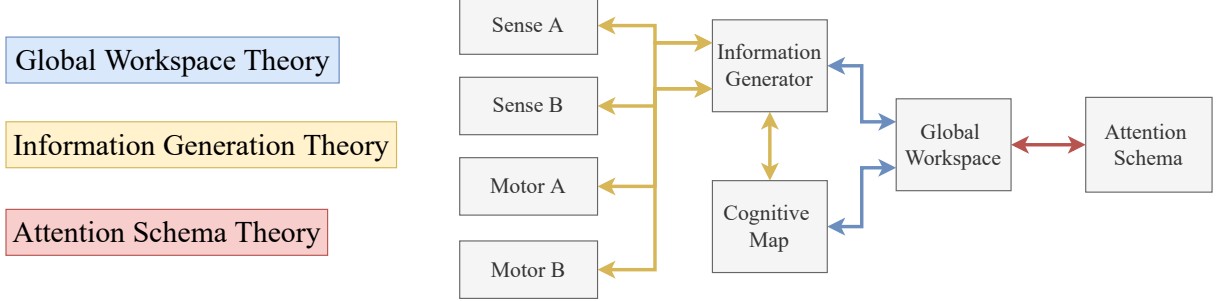

Figure 3: Diagram of how functional theories of consciousness work together to support mental time travel and other intelligent behavior. Sensory and motor information is mediated by a process of internal information generation. This process of information generation is then modulated by a cognitive map, which ensures temporal coherency of generated information. Subsets of this generated information are attended to and thus maintained and manipulated within the global workspace. The content and dynamics of the global workspace are attenuated by an attentional schema modeling those same dynamics.

Other recent work has also attempted to synthesize various functional theories of consciousness. Within the literature surrounding AST, efforts have been made to describe how AST, GWT, and HOT could be resolved into a single theory of conscious function (Graziano et al., 2020). Our proposal to interpret AST and GWT as forms of higher order thought and to propose computational mechanisms for these theories can be seen as an extension of this line of thinking. A similar attempt has been made in the global workspace literature, with Dehaene et al. (2017) proposing that information processing in the brain can be divided into unconscious processing as well as two different kinds of conscious processing. In their taxonomy, the first kind of conscious processing corresponds to the global workspace, while the second corresponds to various forms of meta-cognition, which we have treated here under the umbrella of AST. Separately, Safron (2020) has proposed the Integrated World Model Theory (IWMT), which attempts to reconcile GWT, IIT, and Active Inference (Friston, 2010) (a theory related to IGT). This independent formulation of a synthesis between these theories is largely compatible with the proposals of this work, as it likewise suggests a system for experience generation, and for higher-level abstract representation and information sharing. While IWMT proposes specific candidate architectures such as folded variational autoencoders and graph neural networks,

we explore a broader range of possible computational implementations of the systems described here. Below we explore the space of these possibilities in greater depth.

## 7.2 Implementing access consciousness in artificial agents

We expect that agents will require a cognitive architecture which combines all three functional theories of consciousness described here in order to perform mental time travel in environments with comparable complexity to the "real world." If such agents were developed, they would consequently exhibit greater generalization compared to many current AI systems, as measured by speed in learning novel skills with little experience or prior knowledge. Here we turn to the question of what research directions may help to fully realize this capacity. We utilize as our guide GWT, IGT, and AST, and explore the benefit that their interplay has for mental time travel and generally intelligent behavior more broadly.

In order to perform mental time travel, artificial agents need a past, present, and future to attend to. These can be made possible for the agent through the kind of generative model described by IGT. This functionality can be broken into at least three sub-components. First, there is the capacity to generate internal information which is grounded in the external world. Second, there is the ability to ensure internally generated trajectories are spatially and temporally coherent, such that they likewise correspond to the true world dynamics. Third, there is an episodic memory, which can constrain generalization to be relevant to the agent's experience (Suddendorf & Corballis, 2007). This memory can be interpreted as a generative model which assigns and maintains high probability to relevant seen events in a first-person perspective. There has been significant progress made in the fields of generative modeling, model-based RL, and memory-augmented architectures, which correspond to these capacities.

Research in generative models has progressed rapidly in the last decade, with unconditional models capable of creating ever more realistic images (Dhariwal & Nichol, 2021; Casanova et al., 2021) and audio (Kong et al., 2020; Goel et al., 2022). With extra information, conditional models can generate even more impressive samples: for example, exhibiting compositionality in free-form text-to-image generation (Nichol et al., 2021). Generative models trained on large amounts of static data (Radford et al., 2021) are able to generalize to such an extent that they can be used to create flexible vision-and-language-conditioned policies for robotic manipulation in the real world (Shridhar et al., 2022). While scaling datasets and model architectures has worked well, more structured approaches could lead to more data-efficient generalization: for instance, generative models with explicit machinery for counterfactual reasoning (Sauer & Geiger, 2021). Although IGT does not explicitly require (Pearl) causality (Pearl, 2009), given the role of causality in human mental models (Waldmann & Hagmayer, 2013), as well as its role in OoD generalization (Arjovsky et al., 2019), we expect it to play a role in the construction of generative models that can be used for mental time travel. In particular, as counterfactual reasoning is based around manipulating individual observations, it can be effectively paired with episodic memory.

In the brain, the hippocampus has been linked with the generation of spatially and temporally coherent sequential trajectories (Hassabis & Maguire, 2009; Behrens et al., 2018). The dynamics of it and other interconnected brain networks remains a source of inspiration for research into implementations of functionally similar cognitive maps in artificial agents (Whittington et al., 2022). One recent work along these lines combines a Hopfield network with a recurrent neural network to enable rapid storage and retrieval of observations in novel structured environments (Whittington et al., 2020). The authors furthermore demonstrate that their model develops a number of classical hippocampal-like representations, such as place, grid, and border cells. Additional recent work based heavily on abstract models of the hippocampus demonstrated graph-based models capable of learning the underlying structural dynamics of environments in a few-shot setting (George et al., 2021).

Model-based RL methods have also improved in their capabilities, becoming a strong alternative to model-free methods (Schrittwieser et al., 2020; Hafner et al., 2020). When combined with search algorithms such as MCTS, which can explore novel, task-relevant trajectories, such methods are capable of preplay-like behavior. A study with one state-of-the-art method (Schrittwieser et al., 2020) has indicated that planning and the self-supervision involved in learning a world model can aid generalization, although current methods are still limited in this capability (Anand et al., 2021). Again, incorporating further structure into the model can aid

generalization, by for example disentangling dynamic and static components (Kim et al., 2020), or learning causal models (Buesing et al., 2018; Gasse et al., 2021). To approach the flexible and context-dependent modeling observed in animals (Clayton et al., 2003), we believe it is important to focus on the development of models that are capable of reasoning beyond veridical timescales. There are multiple approaches to this: for example, by focusing on bottleneck states (Neitz et al., 2018; Jayaraman et al., 2018). Most pertinent to our thesis is the work of Zakharov et al. (2021), who use episodes to train a "subjective-timescale" model, focusing model capacity on the most salient events to the agent, using the agent's own prediction errors to determine saliency.

Standard recurrent neural networks (Elman, 1990; Hochreiter & Schmidhuber, 1997) can be likened to short term memory, where storage and computation are performed in-place. In the last decade there has been a renewed attempt to decouple memory and compute (Graves et al., 2014; Sukhbaatar et al., 2015), with the aim of enabling longer-term memory storage. While such approaches are promising for enabling AI in the real world, of particular relevance to mental time travel is the development of episodic memory, where events specific to the agent's experience are stored. Initial attempts to imitate hippocampal episodic control (Lengyel & Dayan, 2007) in AI agents used non-parametric (Blundell et al., 2016) and semi-parametric (Pritzel et al., 2017) memories (which store state information from every timestep), enabling rapid learning when compared to standard deep RL algorithms. Scaling such memory is an open challenge, with possible solutions including more sophisticated storage/forgetting mechanisms, compression (Agostinelli et al., 2019), and hierarchy (Lampinen et al., 2021). Still, these algorithms only correspond to replay, while combining episodic memories with generative models could lead to further abilities, such as planning to find outcomes that are of specific relevance to the agent (Zakharov et al., 2021).

Internally generated trajectories of experience are only one piece of information that could be taken into account by an agent at any moment. The agent must balance incoming sensory signals, information generated internally, and outgoing motor commands. A natural way to do this is to aggregate information in a central area, with modality-agnostic representations. Not only do multimodal models have an advantage at learning more abstract concepts (Goh et al., 2021), but they are also necessary for embodied intelligence in the real world. Advances in datasets, architectures, and compute are now enabling the development of architectures and task setups that can enable general-purpose, modality- and task-agnostic models (Jaegle et al., 2021; Wang et al., 2022), which could lay the foundations for agents that learn high-level concepts grounded in real world experience.

The agent then also needs the ability to selectively focus on some pieces of information and ignore others. This is the capacity to attend, and forms the basis of multiple popular theories of consciousness (Posner, 1994; Ganeri, 2017). In machine learning, the Transformer architecture (Vaswani et al., 2017), with a "self-attention" mechanism (Bahdanau et al., 2014), has been extended to a variety of domains (Dosovitskiy et al., 2020), and is core to many modern multimodal architectures (Jaegle et al., 2021; Wang et al., 2022). The ability to perform data-dependent adaptive processing on differently-structured inputs gives models with this mechanism a large degree of flexibility. They have also shown impressive scaling abilities (Brown et al., 2020; Radford et al., 2021). A weakness of the original attention mechanism is its quadratic complexity, but addressing this is an active area of research (Kim & Awadalla, 2020; Choromanski et al., 2020). That said, there are also other approaches to creating attention mechanisms, such as "hard attention" (Schmidhuber & Huber, 1991; Xu et al., 2015), and top-$k$ sparsity (Makhzani & Frey, 2013; Ahmad & Scheinkman, 2019). These approaches have more favorable computational complexity, but make credit assignment more difficult. A prominent line of related research lies in solving the routing problem in mixture-of-expert networks (Shazeer et al., 2017; Lepikhin et al., 2020; Fedus et al., 2021), which would enable more robust and flexible, modular, models.

Together, (input and output) attention and maintenance of information make the core functions of the global workspace (Mashour et al., 2020). As it is proposed to be located in the prefrontal cortex, which is capable of learning abstract rules and performing planning (Russin et al., 2020), we can characterize the global workspace as a working memory, where conscious reasoning occurs over high-level, multi-modal, semantic concepts. In particular, the global workspace has been linked to Kahneman (2011)'s "system 2" deliberative processing (Bengio, 2017; Goyal & Bengio, 2020). Given the limited capacity and focus of the global workspace, as well as the role of verbalization in reporting awareness, Bengio (2017) interprets the

workspace as an area which contains and manipulates sparse factor graphs that represent the relationships between a small set of high-level concepts, extracted from the unconscious. Goyal & Bengio (2020) extends this interpretation to incorporate aspects of embodied intelligence, with a focus on the role of causality within the factor graphs. As described in Subsection 5.6, the ability to deliberately manipulate causal factor graphs allows agents to generalize to novel, OoD scenarios, which is key to intelligent behavior in the real world.

The broad concept in GWT of having information that is kept both independent and shared as necessary has already inspired several novel neural network architectures (Goyal et al., 2019; 2021; Juliani et al., 2022). Although less directly inspired by GWT, related works have focused on using meta-learning to perform causal discovery, moving towards the flexible manipulations thought to be possible in the global workspace (Bengio et al., 2019; Ke et al., 2019). Independent work has also proposed a formal computer-science-based definition of a theoretical global workspace in the form of a "Conscious Turing Machine" (Blum & Blum, 2021). This proposal is also consistent with the general computational principles of the global workspace, and in particular advocates for sparse, disentangled, and causal representations of information. We further note that, with an episodic memory, the ability to sparsely represent different aspects of an experience within the workspace facilitates mental time travel, with past and future experiences compartmentalized against the present. The past can inform counterfactual thinking, disentangling prior experience—for example, playing volleyball at the beach and basketball at the gym—with imagined scenarios, such as playing volleyball at the gym.

The phenomenon of mental time travel finally requires that the agent performing the time-traveling be aware of itself doing so. This can be made possible through a meta-level representation of the attentional process at work, as proposed by AST. In this system, the agent represents its own attention over the simulation of internal experience, as opposed to simply that of exogenous experience (Graziano & Webb, 2015). This awareness provides not only an autonoetic self-representation to the agent, but allows the agent to adapt its policy of attention based on the current behavioral context. In cases where this context is imagined, the agent would be able to simulate possible policies, each which respond according to the imagined situation, rather than the current one. This ability to take future goals into account is in line with the "prospective" learning seen in natural agents, as opposed to the largely "retrospective" learning implemented in current AI systems (Vogelstein et al., 2022).

Self-awareness, as conjectured by AST, is not clearly mechanistically defined. As realized by the original authors, AST could be construed as an adaptive attention policy within an RL agent (Wilterson & Graziano, 2021). However, one important feature seems to lie in broader aspects of meta-learning, where the agent is trained in order to adapt quickly to novel situations. Interestingly, evidence has been provided for memory-based meta-learning to be capable of modeling some dynamics of the prefrontal cortex (Wang et al., 2018)—the same region proposed to be the location of various cognitive schemas in humans. Meta-learning is a large and ongoing topic of research, and while meta-RL is of particular relevance to intelligent behavior, we find works on learning higher-level task manifolds also noteworthy for the adaptive component of mental time travel (Zamir et al., 2018; Achille et al., 2019). For the most faithful implementation of AST, we believe a "metacontroller", which adapts where and how much computation should be spent in decision making, will be a key component (Hamrick et al., 2017).

Another important feature of AST is the explicit representation of self. The standard example given is that a person looking at an apple can report having an experience of looking at the apple—which necessitates an explicit self-model (Graziano & Webb, 2015). It has been hypothesized that theory of mind—the ability to model what other agents are thinking (Frith & Frith, 2005)—is heavily entwined with self-awareness (Graziano, 2019). While multi-agent research is a large field (Albrecht & Stone, 2018), systems which try to explicitly model the policies of other agents (Rabinowitz et al., 2018; Foerster et al., 2019) could be an important step towards the developments of agents with models of themselves as well. Alternatively, the development of self-models could naturally precede the emergence of theory of mind faculties (Graziano, 2014).

While GWT has been an inspiration for recent AI architectures (Goyal & Bengio, 2020; Blum & Blum, 2021), the integration of further theories of consciousness should lead to more capable autonomous agents.

The first such system would be a combination of IGT and GWT. In this case, an artificial agent would be able to internally generate coherent sequences of experience, and attend to them in an abstracted way utilizing a global workspace. Recent empirical work has created a simple version of such a system, capable of generalization within grid world domains (Zhao et al., 2021). We would expect that such systems would be able to at least perform a strong form of preplay, and thus could adapt to changes in its environment relatively rapidly and efficiently. Without the additional capacity for a meta-representation of self provided by AST however, the behavioral flexibility with respect to potential future needs of such agents still remains limited, and likewise falls short of mental time travel.

The second system we can imagine is a combination of GWT and AST. In this case, information comes directly from the environment, but can be selectively attended to, abstractly represented, and the attention process can be controlled by a higher level meta-representation of that process. This approach is consistent with a recent proposal to interpret the global workspace as a representational space of possible skills or tasks (a schema space) (VanRullen & Kanai, 2021). Related recent work has also provided an empirical instantiation of this concept by utilizing hypernetworks to generate task-specific networks (Lampinen & McClelland, 2020). We expect that such a system would be able to display rapid learning in the face of novel scenarios, although without the ability to internally generate experience as described by IGT, we expect it would still be orders of magnitude less sample efficient than an agent capable of mental time travel.

All of the three systems described above must work together in order to enable true mental time travel as originally defined by Tulving. This involves agents capable of internally generating coherent trajectories of experience, attending to and manipulating those experiences in an abstract way, and adapting their behavior based on a meta-representation of self. What we find is that in many ways the field of AI is quite far along the path of realizing each of these systems, and even some of their potential pairings. We believe that a deeper understanding of the ways in which these systems can potentially work together can enable the development of agents with more general intelligence. Setting a research goal of implementing true mental time travel serves as one possible way in which this synergy can be encouraged.

## 8 Conclusion

While the quest to understand consciousness in a scientific manner is seen as contentious to many, the "hard question" (as opposed to the "hard problem") of what consciousness is for is amenable to scientific enquiry (Dennett, 2018). In particular, awareness, corresponding to information available for conscious access (Block, 1995) appears to be tied to many useful cognitive abilities. In this work, we reviewed GWT, IGT, and AST, three contemporary functional theories of consciousness, backed by empirical evidence from neuroscience studies, which can each be interpreted from the lens of access. By considering each theory from a common computational framework derived from contemporary AI research, we provided a blueprint by which the systems described in each theory can be unified into a single larger system of conscious access. We motivated this unified system through the example of mental time travel as described by Tulving (2002). In doing so, we aimed to clarify the often ambiguous meaning of mental time travel in the psychology and neuroscience literature, by distinguishing it from the related abilities of replay and preplay. We believe the focus on mental time travel is pertinent to a wide swathe of researchers studying cognitive abilities, as it is not only strongly linked to consciousness, but also to future planning and intelligence (Clayton et al., 2003).

Meanwhile, in recent years the field of AI has produced agents with what appears to be increasingly greater intelligence. Models trained on large amounts of data are able to generate language and computer code that is sometimes indistinguishable for what a human would produce (Brown et al., 2020), or even best humans at complex real-time strategy games (Berner et al., 2019; Vinyals et al., 2019). Despite these successes, most models still fail to count as significantly intelligent when the amount of prior experience and knowledge built into the system is taken into consideration (Chollet, 2019).

As a potentially complementary approach to scaling AI models (Bommasani et al., 2021), there has been a considerable amount of research that continues to explore useful inductive biases to incorporate into artificial agents. These include works on attention-based operations, model-based RL, multi-modal models, and meta-learning, which can be mapped onto the functional theories of consciousness we have discussed. However, many of these lines of research are still developed separately to each other. We hope that by demonstrating

the extent to which these three systems must work together in humans to support mental time travel, it will motivate researchers to develop models which attempt to address not just one or two, but all three of the functional theories of consciousness described here. We believe that doing so is one promising path by which more generally intelligent AI systems may be developed. Likewise, by explicitly relating theories of consciousness to concepts and empirical research in AI, we hope to galvanize researchers already working to understand the nature and role of consciousness. Finding common ground between biological and artificial cognitive abilities facilitates empirical, interdisciplinary studies (Crosby et al., 2019), and we hope that our work can similarly lead to more concrete questions on consciousness, such as formulations of tests for the presence of consciousness in artificial agents (Schneider & Turner, 2017).

With a clearer understanding of the concepts involved, we can return to our motivating question concerning the link between emergent consciousness and intelligence in the AI systems. Is the view that the two are co-extensive correct? We find that we can surprisingly respond in the positive, though with a few caveats. Given the philosophical requirements of phenomenal consciousness, we may never be able to say much definitively about the existence or non-existence of qualia for other beings, especially those that exist only in-silico. On the other hand, as pointed out in the thought experiment on philosophical zombies, whether a generally intelligent agent has phenomenal consciousness may be practically irrelevant (Chalmers, 1996). What matters to the question of intelligence is whether those beings function as if they were conscious. Taking the essence of the theories discuss here, we can say that for a being to be generally intelligent in a way which would make sense to us as humans, it must be capable of learning from possible environments and possible selves by attending to and flexibly manipulating information drawn from the past, present, or future, and finally acting on that information in the acquisition of new skills. If that is the case, then we believe that such an agent might indeed be largely functionally consistent with the only beings we know with certainty are conscious: ourselves.

### Acknowledgments

We would like to thank Leonardo Barbosa, Hiro Hamada, Andrew Cohen, Laura Graesser, Yoshua Bengio, and our anonymous reviewers for their helpful feedback on earlier versions of this text. This work was supported by JST, Moonshot R&D Grant Number JPMJMS2012.

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
