# OpenReview forum: "On the link between conscious function and general intelligence in humans and machines"
_TMLR — Accepted by TMLR_

### Review · Reviewer_e4NR · 2022-06-06

**Summary Of Contributions:**

At a high level, the paper argues that incorporating elements of how consciousness is thought to operate might greatly increase the intelligence (defined, following Chollet, as the ability to quickly acquire novel tasks with little experience or priors) of artificial agents, by enabling them to perform certain domain-general cognitive behaviors, such as mental time travel.

The paper explicitly avoids discussion of "phenomenal" consciousness or "qualia", focusing instead on "access consciousness", defined as the contents of conscious awareness, and presumably (although not explicitly stated) the way this content is thought to be generated, selected and/or manipulated.

Three main views of the way consciousness operates (called, somewhat confusingly, "functional consciousness") are presented: Global Workspace, Information Generation, and Attentional Schemas.

Chollet's definition of intelligence (quick acquisition of truly novel tasks with little priors or experience) is endorsed, and shown to be superior in humans than in animals and, a fortiori, machines (though the main example chosen, i.e. language, may be somewhat unfortunate - see below).

Machine learning approaches to such "general" intelligence are discussed, and found wanting (for good reason).

The authors introduce mental time travel as an example of a domain-general cognitive ability that seems to be of great benefit to our "intelligence" as defined above, and also seems to require machinery similar to that posited in theories of consciousness previously discussed.

Interestingly, the authors propose a useful distinction between such "mental time travel" and the phenomena of direct experience, replay and preplay (though some clarifications might be useful, see below).

It is argued that human ability for mental time travel is at the very least vastly superior to that of animals (again, some of this might be clarified).

In what is evidently the main payload of the paper, the authors then propose that the three previously discussed theories of how consciousness operates might all be integrated into a common machinery, which would support the ability to perform mental time travel (and support general intelligence). They also make (quite abstract) suggestions as to how these might be implemented in artificial agents.

The authors conclude that implementing "access consciousness" (again, apparently defined as the way the contents of conscious awareness are selected, generated and / or manipulated) might be an important building block for truly intelligent agents.



**Broader Impact Concerns:**

The paper does not seem to raise broader impact concerns.

**Requested Changes:**



- "Access consciousness" is defined in p. 4, paragraph 3 as the "set of information that is or is not accessible to conscious experience at any given moment".

But in the very next paragraph, it is stated that philosophical Zombies (who, by definition, do *not* have a "conscious experience") are likely to have "access consciousness"!

As written out, this is a plain contradiction! Now, having read the whole paper, I understand that the authors do not actually define "access consciousness" by whether a conscious awareness perceives it. IIUC, they seem to employ "access consciousness" to mean information that is available for report and cognitive reasoning (by whatever controls the agent), in the way that the various models described in section 3 seek to describe. As such, it is indeed available to non-phenomenally-aware entities.

It might be useful to include, in p. 4 paragraph 3, Ned Block's original short definition: "availability for use in reasoning and rationally guiding speech and action". I believe it could have saved me a bit of confusion! For example, it might make it clearer that "access conscousness" denotes a *property* of the information with regard to the agent, rather than denoting the information itself (as the authors' own definition, as quoted above, seemed to imply), or the entity doing the perceiving or the selecting.


- While Global Workspace Theory and Information Generation Theory are reasonably well described, it is difficult to understand the description of "Attentional Schema Theory". Terms like "meta-representations" should at the very least be explained. Also, the work of Koch and Tsuchiya suggests that top-down attention is "buggy" - what is deliberately attended to is not always what enters awareness; it is not obvious how this relates to the discussion at hand. Some clarification of what exactly AST proposes (essentially, rewriting section 3.3) would be very helpful.

- The formal distinction between replay, preplay, and mental time travel is interesting but requires some clarification, especially for the distinction between preplay and MTT. The authors suggest that MTT involves "possible" (i.e. unseen) environments and tasks. But it is not immediately obvious how an agent could have a "first-person experience" of an environment that it has never experienced. It would be very helpful to have some examples of such cases.

- Relatedly, regarding the unique human nature of MTT, the authors might want to mention  the phenomenon of "vicarious trial and error" (Tolman 1948 Psychological Review 1948), which is often interpreted as evidence of something that looks very much like "mental time travel" in rats. Explaining why this phenomenon is or is not MTT might help clarify the concept.

- Using language as an examples of a task that humans can learn with little "prior" (e.g. in section 4.1) is somewhat confusing, since it is widely believed that humans have indeed evolved specific language-learning abilities, separate from our general-learning abilities.

- Minor things: In p. 4, last paragraph: "may not" is probably "might", and in p. 7, last paragraph: "multi-model" is probably "multi-modal" ?

**Strengths And Weaknesses:**



- Strengths:

The paper assembles an immense amount of information over multiple fields of research. The problem being tackled is clearly important. Once the authors' concepts and terminology are understood (which does take a while), the overall argument seems convincing.


- Weaknesses:

Perhaps due to the massive scope of the work, the exposition is occasionally unclear, and certain basic concepts might be described more precisely - especially the concept of "access consciousness", upon which the article rests. See "suggested changes" below.


- Not-necessarily weakness:

The paper is entirely conceptual and prospective. No actual machine learning occurs, and no experiments or hard theoretical results are reported. The entire paper consists of a high-level description of how existing theories of consciousness might be combined to support mental time travel, and presumably other forms of intelligent behavior: it is essentially a massive review-cum-position paper, supported by a thought experiment. The proposed overall model seems promising, but in the absence of experimental results its supposed benefits remain speculative. However, the discussion is undoubtedly very interesting, informative and thought-provoking. Whether this type of paper is appropriate for TMLR is a question I'm happy to leave to the Editor.

---

### Review · Reviewer_U1ps · 2022-06-08

**Summary Of Contributions:**

This paper summarizes three well-known theories of access consciousness from cognitive neuroscience (global workspace theory (GWT), information generation theory (IGT), and attention schema theory (AST)), and argues that these theories and their linkages could aid the development of artificial general intelligence (using Chollet's definition of intelligence as being the ability to rapidly learn new skills with little data or pre-wired domain knowledge). They specifically use the example of "mental time travel", arguing that it is a key feature of intelligence per this definition, and one which demands precisely the resources that GWT, IGT, and AST postulate as being key to access consciousness. Accordingly, they argue that AI could benefit from considering these theories of access consciousness.

**Broader Impact Concerns:**

I do not have any concerns about this.

**Requested Changes:**

- Make note of the fact that Chollet's definition of intelligence is potentially problematic for those who place a lot of emphasis on evolutionary priors.

- Update the discussion of preplay to note that some "preplay" is likely not simulations of future trajectories, but a reflection of recurrent connectivity motifs in the hippocampus.

- Make a clear, more concrete statement as to the concrete direction AI researchers should take given these ideas.

**Strengths And Weaknesses:**

Strengths:

- The paper is clearly written, and was enjoyable to read.
- The thesis is novel, and I think many readers would be interested by it.
- The citations are thorough, and the authors manage to span the disciplinary gap remarkably well.

Weaknesses:

- There is a tension in Chollet's definition of intelligence that the authors don't discuss, namely, what if humans are actually quite dependent on some evolutionary priors for their ability to learn rapidly? Consider, humans aren't fast at learning *everything*, and there are many tasks we may not be able to learn without a large amount of data, if at all (for example, reading QR codes). Moreover, some people argue that AI needs a lot more in the way of structural priors. All that being said, I think humans clearly are good at learning a remarkable range of things rapidly, and often do use their conscious processing to bootstrap that process, so I am happy with the definition. Nonetheless, I think it is worth simply noting in the paper that the definition as provided may be in tension with accounts of animal and artificial intelligence that place a much stronger emphasis on structural priors.

- The account provided of "preplay" in the hippocampus in the paper is a tiny bit confused. Specifically, there are two different forms of preplay that the authors mix here (in fairness, so does the literature, sometimes). One, examined by researchers like Foster, is more in-line with what the authors are talking about, e.g., sweeps forward of potential trajectories in the environment, particularly at choice points. But the other kind, studied by researchers like Dragoi, instead refers to the fact that sequences of activity observed in a novel environment reflect the statistics of sequences observed prior to exposure to that environment. This likely reflects the underlying recurrent connectivity of the hippocampus, and is not a form of simulation of potential trajectories.

- It would be nice if the authors in the abstract and/or conclusion were willing to make a slightly more definitive declaration, e.g., that AI researchers should actively try to build systems that can engage in mental time travel.

---

### Review · Reviewer_bfZw · 2022-06-18

**Summary Of Contributions:**

This paper evaluates the potential link between AI and three prominent theories of consciousness from cognitive neuroscience:global workspace theory (GWT), information generation theory (IGT), and attention schema theory (AST). The paper considers the range of functions addressed by each theory, and proposes a high-level integrative architecture incorporating the major aspects of each theory. There is also an interesting proposal to treat the capacity for 'mental time travel' as a kind of benchmark for consciousness in artificial systems, together with arguments for the importance of each the three major theoretical perspectives in enabling this capacity.

**Broader Impact Concerns:**

I do not envision any ethical concerns arising from this work.

**Requested Changes:**

I think the paper would be improved if the authors address the specific issues raised above. However, I think the more significant issue is whether the review / perspective format is appropriate for TMLR. One potential direction for making this work more suitable to TMLR would be to formalize the 'mental time travel' proposal as a concrete benchmark. I think that could be a useful way to drive the development of the kinds of integrative systems that the paper envisions. Of course, that would also be substantially beyond the scope of the present work.

**Strengths And Weaknesses:**

I found the synthesis of ideas presented in the paper to be generally compelling, and the specific proposal to focus on mental time travel as a benchmark is interesting. The combined review of concepts in both consciousness studies and AI research will also likely be mutually beneficial to both communities. My primary concern is that the paper is largely a review and perspective piece. The high-level proposals concerning integration of multiple theoretical perspectives on consciousness, and specific proposal to treat mental time travel as a benchmark, are interesting, but these are ultimately difficult to evaluate without concrete models or experimental results. I do think that integrative perspective pieces like this can be useful, but I am not certain whether TMLR is the appropriate venue for publication. This is in large part due to the fact that TMLR is a new venue, and I'm not certain what the intended range of article formats is. I took a look at the 'Scope' section of the submission guidelines, and did not find any descriptions matching this kind of article. I will leave it up to the editor to decide whether TMLR is an appropriate venue, but I have also included my specific responses to the issues raised in the article below.

### Preliminaries on philosophy of consciousness, access vs. phenomenal consciousness, etc.
* The article presupposes a particular philosophical view of consciousness that is far from agreed upon by consciousness researchers. The background section on philosophical approaches to consciousness discusses ideas such as the hard problem of consciousness, qualia, access vs. phenomenal consciousness, etc. as though these are uncontroversial philosophical perspectives on the topic. The article furthermore casts the three major theories considered (GWT, IGT, AST) as theories of access consciousness (not phenomenal consciousness), giving the impression that the proponents of these theories endorse this characterization. This is arguably not the case for global workspace theory, and certainly not the case for the attention schema theory. For instance, in a highly influential paper on global workspace theory from Dehaene and Naccache [1], the authors explicitly question the distinction between access and phenomenal consciousness (in the section entitled 'qualia and phenomenal consciousness'), and suggest that both may be part of the same underlying phenomenon, which GWT may account for. AST is even more explicit about this, arguing that the brain does not actually have the properties associated with phenomenal consciousness [2]. I think it is appropriate for the authors to discuss these ideas, and to propose that these three theories be viewed as theories of access consciousness, but it is important to be clear that this is their own proposal, and to also describe alternative philosophical perspectives, particularly as it pertains to the proponents of the three major theories that are discussed. I think this is especially important given that part of the presumed audience for this paper will be researchers in AI who are unfamiliar with work on consciousness -- it is important to provide an accurate and balanced perspective of the field.
* A more theoretically neutral characterization of these theories may be to describe them as 'functionalist' theories of consciousness. This term does not presuppose any theoretical commitments about whether there is something above and beyond functionality (as e.g. the access / phenomenal distinction does), and I think it adequately captures the theoretical scope that the authors are trying to articulate in this section. This would also fit nicely in the discussion of Dennett's 'hard question', which is primarily an attempt to focus discussions about consciousness on questions of functionality.

[1] Dehaene, S., & Naccache, L. (2001). Towards a cognitive neuroscience of consciousness: basic evidence and a workspace framework. Cognition, 79(1-2), 1-37.

[2] Graziano, M.S.A. (2019). We are machines that claim to be conscious. Journal of Consciousness Studies , 26, 94-104.

### Background on theories of consciousness
* There are multiple statements that AST emphasizes the prefrontal cortex (PFC) as the primary brain region involved in computing the hypothesized attention schema. This is not correct. The primary brain region emphasized by the attention schema theory literature is the temporoparietal junction (TPJ) [3], though certainly the PFC may also play an important role.
* The description of IGT states that only the contents of the brain's generative models are accessible to consciousness (not the direct sensory input). How is this distinct from the 'local recurrence' view of consciousness [4], according to which top-down, recurrent processes (presumably implementing a generative model) are necessary for consciousness?
* The proposed role of the attention schema in 'mental time travel' is to provide awareness of the fact that the subject is imagining rather than directly perceiving. The authors should also discuss the 'perceptual reality monitoring' theory [5], which is primarily about this exact capacity.

[3] Webb, T. W., Igelström, K. M., Schurger, A., & Graziano, M.S.A. (2016). Cortical networks involved in visual awareness independent of visual attention. Proceedings of the National Academy of Sciences, 113(48), 13923-13928.

[4] Lamme, V. A., & Roelfsema, P. R. (2000). The distinct modes of vision offered by feedforward and recurrent processing. Trends in neurosciences, 23(11), 571-579.

[5] Lau, H. (2019). Consciousness, metacognition, & perceptual reality monitoring.

### Background on intelligence
* The paper characterizes intelligence as, roughly, the ability to rapidly acquire novel skills. I think this fits with a large body of work in psychology, but the only work cited to support this view is the paper from Chollet [6] proposing a benchmark for AI. It would be good to also mention some of the psychological literature on this topic (e.g. [7]) -- Chollet's conception seems to align fairly well with the psychological notion of 'fluid' (as opposed to 'crystallized') intelligence.

[6] Chollet, F. (2019). On the measure of intelligence. arXiv preprint arXiv:1911.01547.

[7] Cattell, R. B. (1987). Intelligence: Its structure, growth and action. Elsevier.

### Background on generalization in AI
* This section felt a bit too much like a laundry list of various approaches in AI, without a clear connection to the primary proposals of the paper. It would help to include some hints as to how these approaches are ultimately relevant to the three theories of consciousness under consideration, and the capacity for mental time travel.

### Mental time travel
* The description of mental time travel sounds almost synonymous with 'imagination'. Is there any distinction between mental time travel and imagination?
* What is the relationship between mental time travel and general intelligence, as the authors have defined it? The earlier sections of the paper emphasized the importance of few-shot and zero-shot generalization for intelligence, but this is not clearly tied to the capacity for mental time travel.

### Minor notes
* The sentence 'we do not however totally discount the possibility that the study of phenomenal consciousness may not provide functional insights into animal behavior' seems to have one too many negatives?
* ‘The global workspace theory proposed by is just such a…’ - is there supposed to be a citation here?

---

### Author Response · Authors · 2022-06-27
**Revised paper uploaded**

Dear reviewers, editors, and public,

We have now uploaded a revised draft of the paper. This revision is based on our responses to the helpful feedback of each of the three reviewers (whose content is described in each of our responses to the reviews themselves), along with other feedback gathered during the review period. We hope that this new draft addresses the bulk of the actionable concerns presented by the reviewers. Again we thank the reviewers for their careful attention to the concepts presented in the paper, and believe that this new draft presents a stronger and clearer set of ideas as a result of their valuable feedback.

---

### Comment · Action_Editors · 2022-06-30
**Revision uploaded**

Dear Reviewers,

thank you for your helpful reviews. The authors have posted comments to them and uploaded a revised version. Do you think they addressed your suggestions?

best

---

> ### Comment · Reviewer_e4NR · 2022-06-30
> **OK with changes**
>
> Having read the author's reply and the corresponding passages in the revised version, I am happy with the changes and I believe they address my concerns.
>
> Please just be sure to check for typos in the new version, e.g. p.7 section 3.3: "Taking an exampled" (there may be others).
>
> If the editors are willing to accept a purely conceptual paper for TMLR, I do not have any objection with this paper being accepted.

---

> > ### Author Response · Authors · 2022-06-30
> > **Typographical Checks**
> >
> > Thank you for pointing out the typo. In light of it, we have submitted the article to an additional spelling and grammar check, and fixed a number of small issues which arose. The new revision has just been uploaded.

---

### Decision · Action_Editors · 2022-07-15

**Recommendation:** Accept as is

**Comment:**

The manuscript discusses potential links between artificial intelligence and the three main theories of consciousness. It provides an extensive review of these theories, combining concepts from consciousness studies and AI. It proposes an integrative architecture that incorporates the major aspects of each. This architectures is however only described on a high level. The authors furthermore propose to use mental time travel as a benchmark for consciousness in artificial systems.

Strengths:
- All reviewers agree that the presented ideas are very interesting and inspiring.
- The proposal is novel
- The manuscript is well written and nice to read.
- The extensive review part is inspiring and cross-disciplinary, a survey certificate is recommended.

Weaknesses:
- The main weakness is that the paper is more of an opinion/perspective type. All proposals are interesting, but there is no hard evidence for them. Rather, the arguments are more on a philosophical/thought-experiment level. No simulations or other experiments are performed.

Conclusion:
Although the paper does not present hard evidence, we believe that it is still a convincing work, thought provoking and potentially influential for future AI systems. I therefore recommend acceptance.